# S-cone photoreceptors in the primate retina are functionally distinct from L and M cones

Jacob Baudin[1,2,3], Juan M Angueyra[1,2], Raunak Sinha[1,2,4]*, Fred Rieke[1,2]

[1]Department of Physiology and Biophysics, University of Washington, Seattle, United States; [2]Howard Hughes Medical Institute, University of Washington, Seattle, United States; [3]Google Inc., Seattle, United States; [4]Department of Neuroscience, University of Wisconsin School of Medicine and Public Health, Madison, United States

**Abstract** Daylight vision starts with signals in three classes of cone photoreceptors sensitive to short (S), middle (M), and long (L) wavelengths. Psychophysical studies show that perceptual sensitivity to rapidly varying inputs differs for signals originating in S cones versus L and M cones; notably, S-cone signals appear perceptually delayed relative to L- and M-cone signals. These differences could originate in the cones themselves or in the post-cone circuitry. To determine if the cones could contribute to these and related perceptual phenomena, we compared the light responses of primate S, M, and L cones. We found that S cones generate slower light responses than L and M cones, show much smaller changes in response kinetics as background-light levels increase, and are noisier than L and M cones. It will be important to incorporate these differences into descriptions of how cone signaling shapes human visual perception.
DOI: https://doi.org/10.7554/eLife.39166.001

*For correspondence:
raunak.sinha@wisc.edu

## Introduction

Sensory receptors pose fundamental limits to perception. Perceptual sensitivity to flickering lights, for example, is dramatically different at low- and high-light levels, and much of this dependence on light level can be traced to differences in the kinetics of responses of the rod and cone photoreceptors themselves. Under many conditions, perceptual sensitivity to flickering lights is also lower for signals originating in short- (S) wavelength-sensitive cones than for signals originating in middle- (M) or long- (L) wavelength-sensitive cones (*Brindley et al., 1966*; *Green, 1969*; *Smithson and Mollon, 2004*). Responses mediated by S vs L and M cones (hereafter LM cones) differ similarly in higher brain centers (*Cottaris and De Valois, 1998*; *Tailby et al., 2008*). These differences appear to originate before cone signals are combined to create color-opponent pathways, perhaps as early as in the cones themselves (*Lee et al., 2009*).

Several considerations suggest that signaling could differ in S vs LM cones. First, responses of salamander and goldfish S cones are considerably slower than those of L cones (*Rieke and Baylor, 2000*; *Howlett et al., 2017*), and salamander S cones exhibit lower noise than L cones (*Rieke and Baylor, 2000*). Second, the kinetics of responses of primate cones differ across retinal regions, providing a precedent for systematic variations in signaling in primate cones (*Sinha et al., 2017*). Third, the evolutionary split between S cones and LM cones is estimated to have occurred more than 500 million years ago (*Nathans et al., 1986*), providing ample opportunity for differences in signaling to emerge. Although previous recordings from primate cones find similar kinetics across responses of different cone types (*Baylor et al., 1987*; *Schnapf et al., 1990*; *Hornstein et al., 2004*; *Cao et al., 2014*), the small number of recorded S cones in these studies precludes a definitive comparison.

Here, we directly compare response kinetics, sensitivity to background light, and noise properties of primate S and LM cones. By learning to identify S cones based on their morphological features (*Ahnelt et al., 1990*; *Curcio et al., 1991*), we mitigated the challenges associated with their relative scarcity and collected a set of S cone recordings many times larger than those previously used to analyze their response properties. These data revealed that S cones differ from LM cones in kinetics, adaptation and noise. We show further that the differences in cone signaling impact retinal output signals and hence likely contribute to perception.

## Results

The results below are divided into four sections: (1) comparison of the response kinetics of S and LM cones; (2) characterization of how the kinetics of S- and LM-cone responses depend on background-light level; (3) comparison of S- and LM-cone responses in retinal ganglion cells; and (4) comparison of noise in S and LM cones.

### S-cone responses are slower than those of L- and M- cones

#### Peripheral cones

To compare the kinetics of cone responses across spectral types, we recorded voltage responses to brief flashes delivered in the presence of background light that produced matched photon-absorption rates (R*/s) in each cone type. A cell's flash response was taken to be the average response across multiple trials. *Figure 1* shows responses of many individual cones at several background-light levels. To compare response kinetics, we averaged responses across cones of each type at a given background. Two features are apparent in these averaged responses: (1) S-cone responses reach a peak amplitude later than L- or M-cone responses, particularly at the higher background-light levels (*Figure 1B*); and (2) the kinetics of S-cone responses change less with background than those of L- and M-cone responses (*Figure 1C*). We explore each of those issues in more detail below.

*Figure 2A* shows the average responses of peripheral cones at 5000 R*/cone/s; at this light level, S-cone responses reached a peak considerably later than L- or M-cone responses (*Figure 2C*; 45.3 ± 0.9 ms for 36 S cones vs 36.3 ± 0.6 ms for 26 M cones and 36.7 ± 0.6 ms for 49 L cones, mean ± sem). S-cone responses were similarly slower at 50,000 R*/cone/s (*Figure 2D*; time to peak of 44.5 ± 0.9 ms for 28 S cones, 34.1 ± 0.9 ms for 17 M cones and 33.1 ± 0.7 ms for 42 L cones), while responses of different cone types were much more similar at 1000 R*/cone/s (*Figure 1B*, top right; see also section on adaptation below). Analysis of response durations led to a similar conclusion: at both 5,000 and 50,000 R*/cone/s the full-width-at-half-maximum (FWHM) of the S cone responses was substantially greater than that of L and M cones (5000 R*/cone/s, FWHM of 37.5 ± 1 ms for 36 S cones vs 30.2 ± 0.8 ms for 26 M cones and 32.2 ± 0.9 ms for 49 L cones; 50,000 R*/cone/s, FWHM of 42.7 ± 1.1 ms for 28 S cones, 29.4 ± 1.1 ms for 17 M cones and 29.0 ± 1.0 ms for 42 L cones).

*Figure 2A–D* indicates that, on average, S cones had slower responses than L and M cones, and that L- and M-cone responses were nearly identical. One concern is that light responses can vary across cones (e.g. *Figure 1*) and across different retinas, potentially biasing our measured cone responses. To control for such potential bias, we compared S-cone responses with those of LM cones from the same piece of retina. Each light purple point in *Figure 2E and F* plots the average time to peak of all LM-cones recorded in a given piece of retina against the average time to peak of all recorded S cones in the same retinal piece. S-cone responses were distinctly slower than LM-cone responses across 14 retinas at 5000 R*/cone/s and 12 retinas at 50,000 R*/cone/s. Differences similarly persisted when we compared responses of individual S cones to those of LM cones measured immediately before or afterwards in the same retina (not shown).

We used the same procedure to compare L- and M-cone responses, including checking the validity of grouping them as a reference for S-cone kinetics. When we compared the average L-cone time to peak to the average M-cone time to peak in each piece of retina, the pairs clustered close to unity (*Figure 2—figure supplement 1*). At 5000 R*/s, there was no measurable difference in the paired average L- and M-cone times to peak (−2.0 to +2.2 ms, 95% CI), while the difference in the paired average S-cone and LM-cone times to peak was 8.5 ms (+6.4 to+10.6 ms, 95% CI). At 50,000 R*/s, these differences were 1 ms for L vs M cones (−2.0 to +4.2 ms, 95% CI) and 11.5 ms for S vs LM

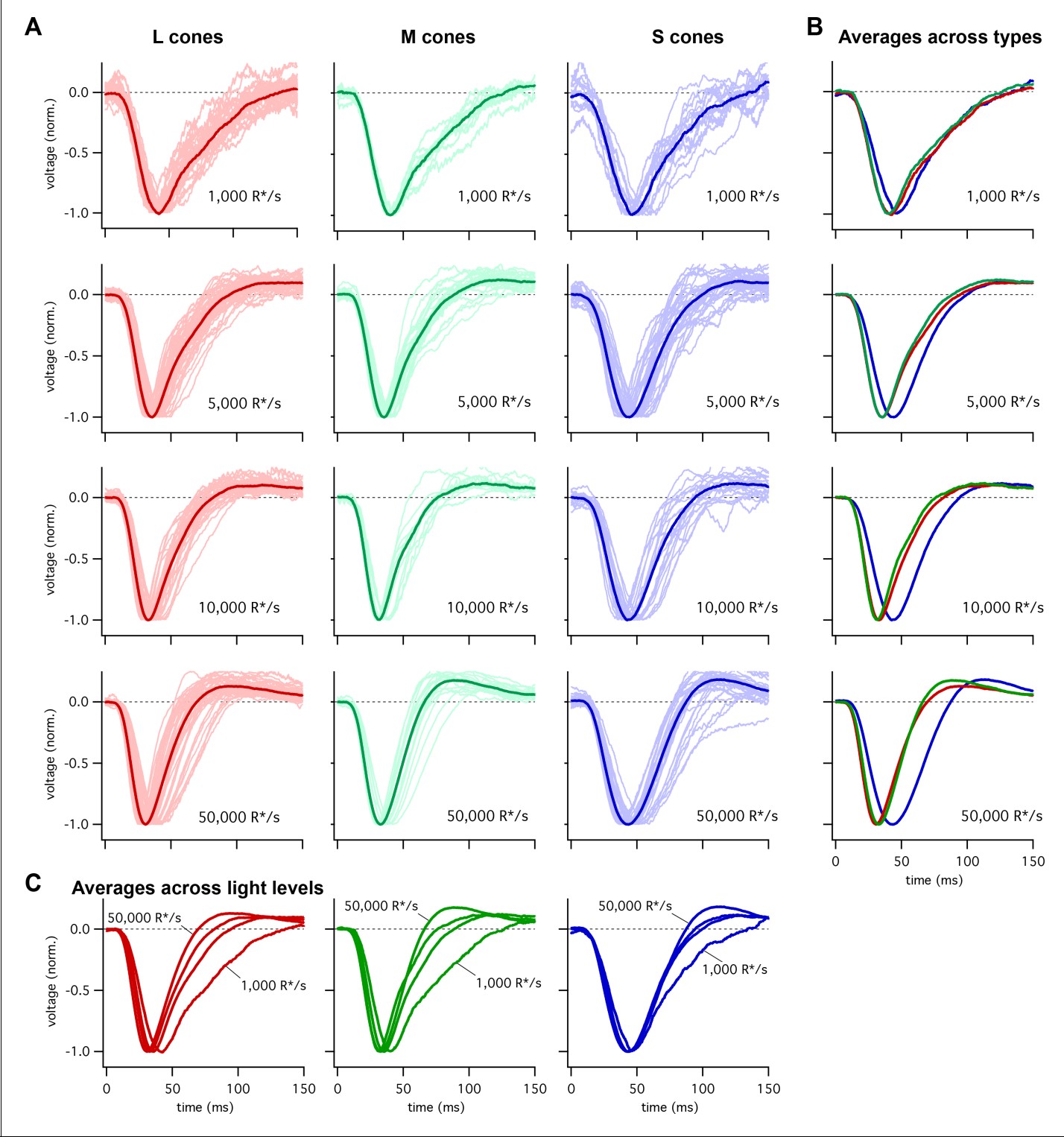

**Figure 1.** Collected flash responses from each recorded peripheral cone across backgrounds. (A) Thin traces show voltage responses of individual cones to a 10 ms flash across several background-light levels. Traces are averages of 5–10 trials, and have been normalized in each cell. Thick traces show averages across cells. Data in this figure is collected from >10 retinas. (B) Superimposed average responses of each cone type, organized by background-light level. (C) Superimposed average responses across backgrounds, organized by cone type.

DOI: https://doi.org/10.7554/eLife.39166.002

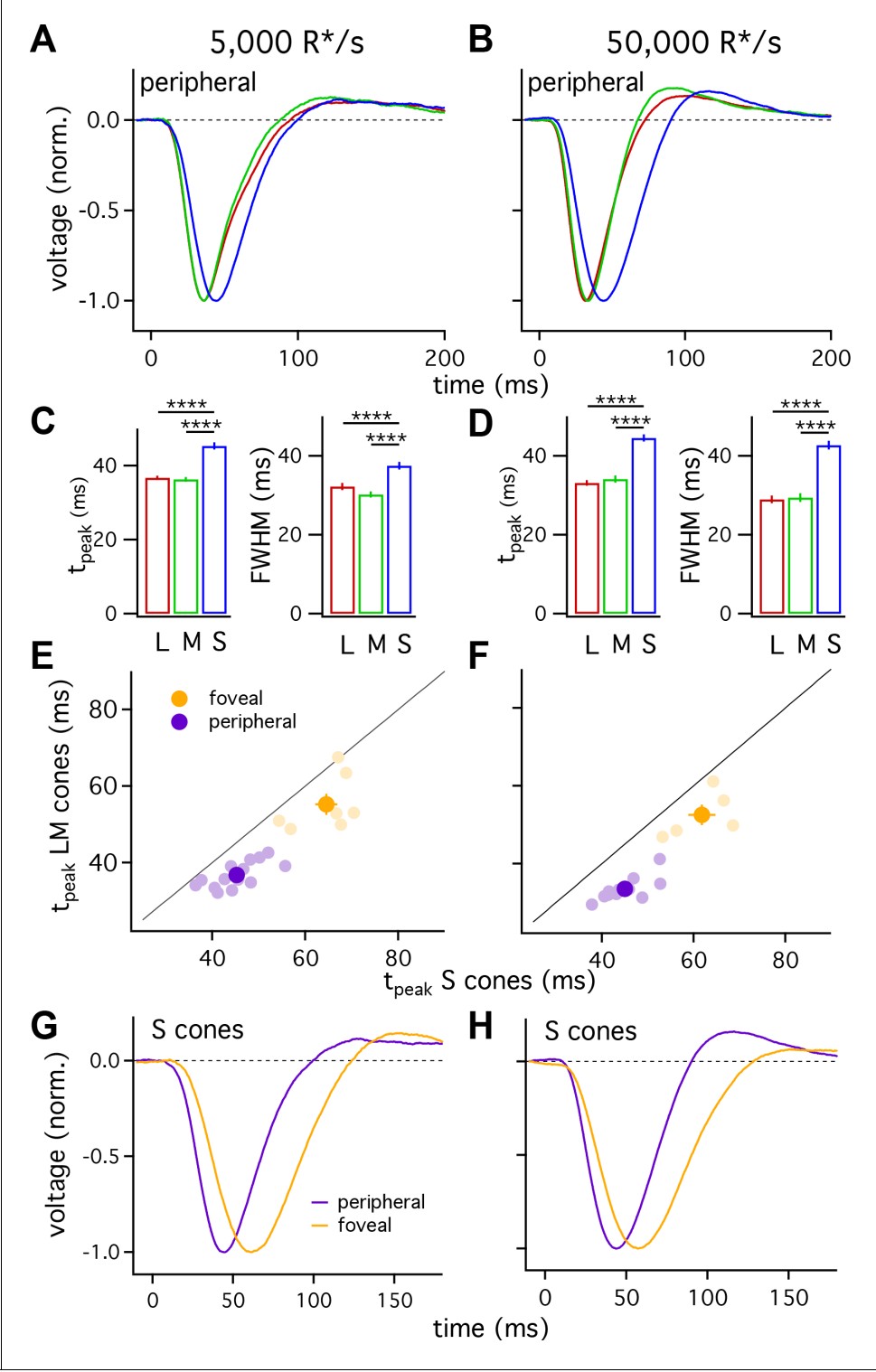

**Figure 2.** S cones are slower than L and M cones across retinal eccentricities. (**A**) Average normalized voltage responses of L, M and S cones on a background of 5000 R*/s. These averages, unlike those in *Figure 1*, include only cells from scatter plots in E. Data in this figure is collected from >10 peripheral retinae and >= 5 foveae. (**B**) As in A for data collected on a background of 50,000 R*/s. Only cells from scatter plots in F are included. (**C**) Times to peak (left) and full-width-at-half-maximum (FWHM, right) across cone types at a background of 5000 R*/s. The mean ± sem times to peak were 36.7 ± 0.6 ms for L cones (n = 49), 36.3 ± 0.6 ms for M cones (n = 26), and 45.3 ± 0.9 ms for S cones (n = 36). p-Values from unpaired t-test. For the same cones, mean ±sem of the FWHM was 37.5 ± 1 ms for S cones, 30.2 ± 0.8 ms for M cones and 32.2 ± 0.9 ms for L cones. **** denotes p<0.0001. (**D**) Times to peak and FWHM as in C at 50,000 R*/s. The mean ± sem times to peak were 33.1 ± 0.7 ms for L cones (n = 42), 34.1 ± 0.9 ms for M cones

*Figure 2 continued on next page*

*Figure 2 continued*

(n = 17), and 44.5 ± 0.9 ms for S cones (n = 28). For the same cones, mean ±sem of the FWHM was 42.7 ± 1.1 ms for S cones, 29.4 ± 1.1 ms for M cones and 29.0 ± 1.0 ms for L cones). (E) Average time to peak of S cones compared to average time to peak of pooled L and M cones in peripheral (purple) and foveal (gold) retina on a background of 5000 R*/s. Each lightly shaded point represents the average time to peak from a single piece of retina (15 peripheral and seven foveal pieces; peripheral cells are those from A and C; foveal data comprises 29 S cones, 24 M cones and 25 L cones). Dark points with error bars represent the mean ±sem across all pieces at a given eccentricity. S cones are significantly slower than their LM counterparts in peripheral ($p<10^{-6}$, paired t-test) and foveal ($p<0.05$, paired t-test) retina. (F) As in E, for data collected on a background of 50,000 R*/s. Peripheral ($p<10^{-6}$, paired t-test) and foveal ($p<0.05$, paired t-test) S cones were significantly slower than their LM counterparts (11 peripheral pieces, five foveas; peripheral cells are those from B and D; foveal data comprises 21 S cones, 8 M cones, and 12 L cones). (G) Average S-cone flash responses from peripheral and foveal retina on a background of 5,000 R*/s. Times to peak of 63.5 ± 1.8 ms for 29 foveal S cones and 45.3 ± 0.9 ms for 36 peripheral S cones. (H) As in (G) on a background of 50,000 R*/s. Times to peak of 59.1 ± 2.1 ms for 21 foveal S cones and 44.5 ± 0.9 ms for 28 peripheral S cones.

DOI: https://doi.org/10.7554/eLife.39166.003

The following figure supplements are available for figure 2:

**Figure supplement 1.** Kinetics of L- and M-cone flash responses are similar.

DOI: https://doi.org/10.7554/eLife.39166.004

**Figure supplement 2.** S and LM cones have similar response shapes Left and right panels superimpose average responses from each cone type with the peak absolute amplitude and the time to peak normalized to one.

DOI: https://doi.org/10.7554/eLife.39166.005

cones (+9.3 to+13.7 ms, 95% CI). Thus, L-and M-cone response kinetics differ minimally whereas S-cone responses are substantially slower than those of LM cones at these light levels.

Do the slower S-cone responses reflect slowed response onset, recovery or both? To answer this question, we normalized the time to peak of the average responses for each cone type. These normalized responses superimposed more closely than the unnormalized responses throughout both response onset and recovery (*Figure 2—figure supplement 2*). Hence the slower S-cone responses compared to LM-cone responses appear to reflect a slowing of both response onset and recovery rather than a preferential slowing of one or the other (see Discussion).

## Foveal cones

L- and M-cone responses are slower in the fovea than the periphery (*Sinha et al., 2017*). To determine whether S cones exhibit similar regional differences in kinetics, we compared responses of S cones in foveal (within 500 µm of foveal pit) and peripheral (>5,000 µm from foveal pit) retina. Similar to foveal LM cones, responses of foveal S cones were slower than those of their peripheral counterparts at background light levels of both 5000 and 50,000 R*/cone/s (*Figure 2G and H*).

Slower kinetics of foveal S cones relative to peripheral S cones do not mandate that foveal S cones will be slower than foveal LM cones. Because of the importance of the fovea for perception, we also directly compared foveal S-cone responses with those of foveal LM cones to determine if the differences in kinetics observed in peripheral retina held in the fovea. S-cone responses, averaged across all cells that we could reference to a L or M cone in the same fovea, were slower than LM-cone responses at both 5000 and 50,000 R*/cone/s (*Figure 2E and F*; S cones, 63.5 ± 1.8 ms at 5000 R*/s and 59.1 ± 2.1 ms at 50,000 R*/s; LM cones, 56.1 ± 1.1 ms at 5000 R*/s and 53.6 ± 1.3 ms at 50,000 R*/s). As for peripheral cones, these differences persisted when responses of S cones were compared to responses of LM cones measured in the same fovea (*Figure 2E* shows results from 7 foveas at 5000 R*/cone/s and *Figure 2F* results from 5 foveas at 50,000 R*/cone/s). These data demonstrate that (1) like LM cones (*Sinha et al., 2017*), responses of foveal S cones are slower than those of peripheral S cones; and, (2) responses of S cones are slower than those of LM cones in both the fovea and periphery.

## S- vs LM-cone kinetic differences hold for periodic stimuli

Many psychophysical studies probing kinetics of S-cone mediated visual signals have used periodic stimuli (*Brindley et al., 1966*; *Marks and Bornstein, 1973*; *Stockman et al., 1991*; *Stockman et al., 1993*). The slower kinetics of the flash response predict that S-cone sensitivity should fall at lower temporal frequencies than LM-cone sensitivity. To test this prediction directly, we measured frequency tuning curves of voltage responses from cones of each spectral type. *Figure 3A and C* show responses of example cones of each type at 5000 and 50,000 R*/cone/s; responses at low- and high-

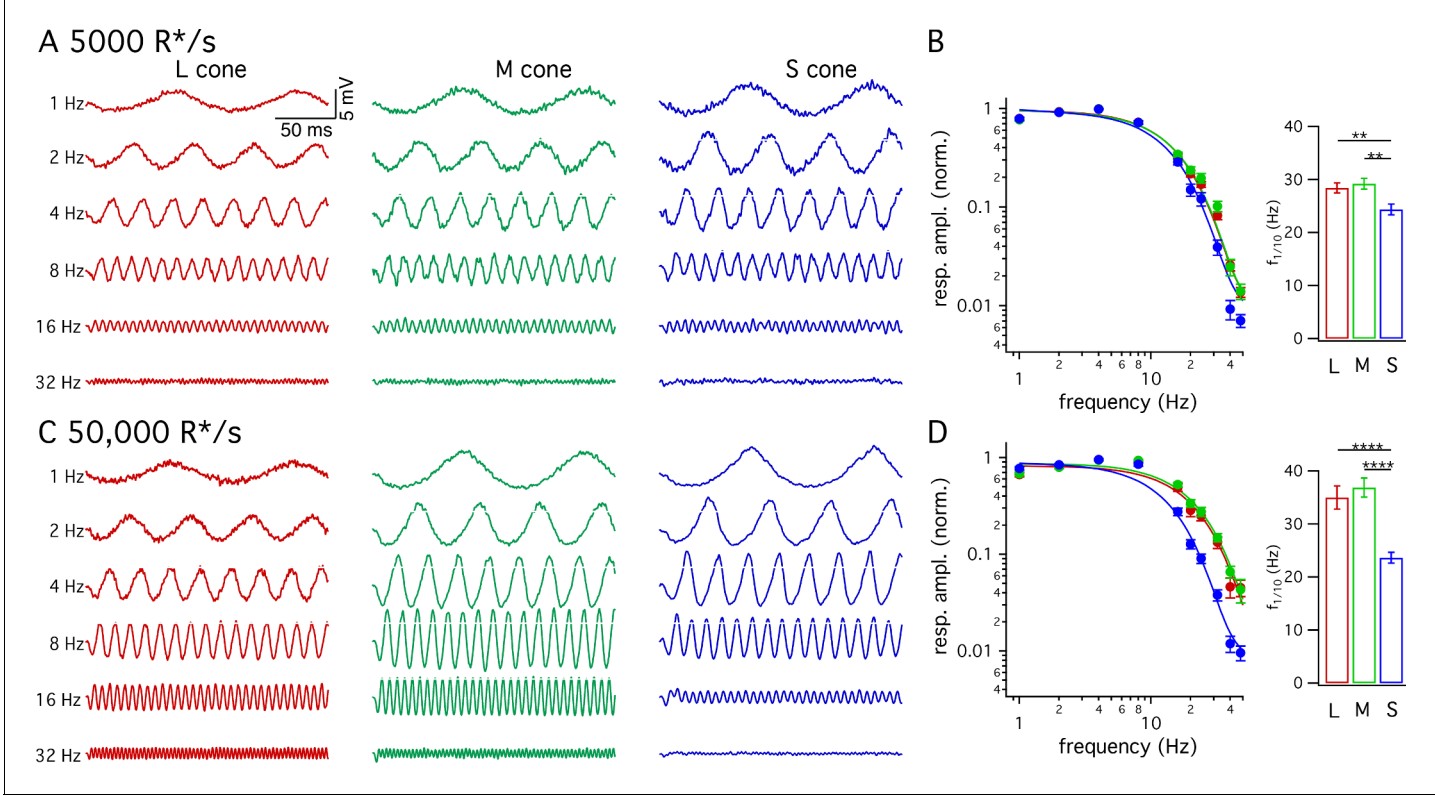

**Figure 3.** Frequency tuning differs for S and LM cones. (A) Voltage responses to sinusoidal stimuli across a range of frequencies from example L, M and S cones. Mean light level 5000 R*/s. Stimulus contrast increased with increasing temporal frequency to avoid saturating responses at low frequencies. Data in this figure is collected from >10 retinas. (B) Frequency tuning curves across cone types. Points with error bars are mean ± sem temporal frequency sensitivities of L, M and S cones on a background of 5000 R*/s. Values are the normalized response amplitudes across frequencies. Curves show best fit of power spectrum of *Equation 1* to the population data. Right panel shows mean ±sem frequencies at which response amplitudes decreased to 10% of their maximum value, which were 28.4 ± 0.9 Hz in L cones (n = 26), 29.2 ± 1.0 Hz in M cones (n = 17), and 24.4 ± 1.0 Hz in S cones (n = 15). P values from unpaired t-test; ** denotes p<0.01 (C) Responses from the same cones and frequencies as A at 50,000 R*/s. (D) As in B for data collected on a background of 50,000 R*/s. Mean ± sem temporal frequencies for 10% maximum gain were 35.0 ± 2.2 Hz in L cones (n = 17), 36.9 ± 1.8 Hz in M cones (n = 21), and 23.6 ± 1.0 Hz in S cones (n = 16). **** denotes p<0.0001.

DOI: https://doi.org/10.7554/eLife.39166.006

light levels are from the same cone of a given type and the stimulus contrast was increased with increasing frequency to enable measurements at high frequencies. *Figure 3B and D* plot the normalized contrast sensitivity (amplitude of the modulated response divided by contrast) as a function of frequency for each cone type. These curves quantify how robustly a given cone type responded to sinusoidal stimuli across a range of frequencies. As expected from the flash response data, the average S-cone tuning curve declines more quickly with increasing frequency than the average LM-cone tuning curve. To quantify these differences, we calculated the frequency at which the tuning curve of each cone type dropped by a factor of 10 from its maximum value (*Figure 3B and D*). This frequency was substantially lower for S-cone responses compared to LM-cone responses. These results confirm that the slower S-cone kinetics observed in responses to brief flashes hold for responses to sinusoidal stimuli.

## S- vs LM-cone kinetic differences originate in phototransduction

The kinetic differences between cone types described above could originate in phototransduction and/or in the conversion of transduction currents to inner segment voltages. A persistence of kinetic differences in voltage-clamp recordings would indicate a large contribution from phototransduction, as is the case in salamander cones (*Rieke and Baylor, 2000*). Restriction of kinetic differences to current-clamp recordings would indicate an origin in the current-to-voltage transformation, consistent

with goldfish cones (*Howlett et al., 2017*) and with the role of HCN-channel activity in speeding photoreceptor responses (*Barrow and Wu, 2009*; *Della Santina et al., 2012*).

To distinguish between these possibilities, we recorded voltage-clamp responses to Gaussian noise stimuli (*Figure 4A*). For each cell, we calculated the linear filter that optimally maps the noise stimulus to the cone's responses (*Rieke et al., 1997*; *Figure 4B*). Each cell's linear filter provides an estimate of its impulse response (i.e. its response to a brief flash of light). Average linear filters for S cones were substantially slower than those for LM cones (*Figure 4B*). These differences held for S

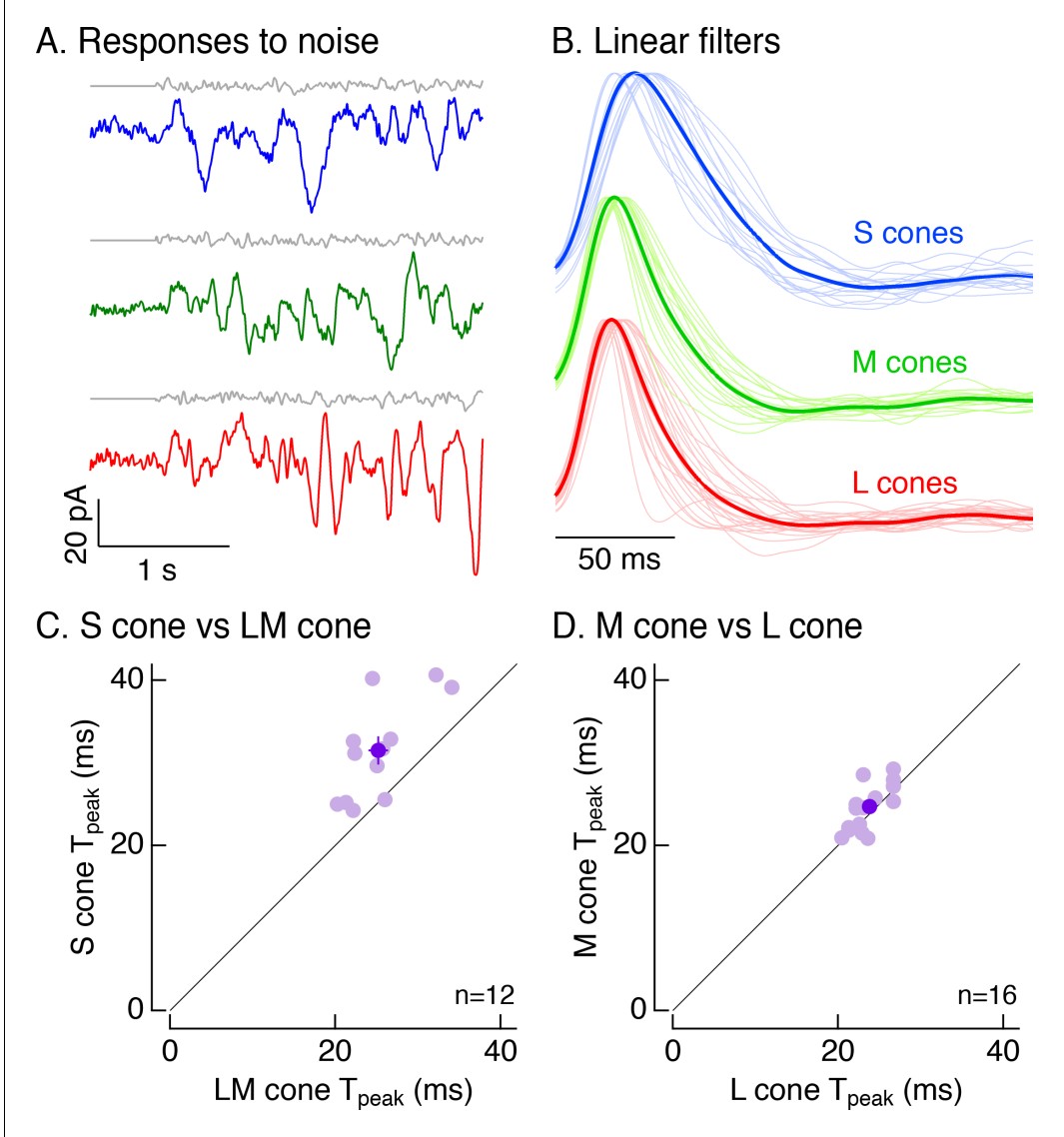

**Figure 4.** Cone photocurrents differ in kinetics. (A) Example current responses of a voltage-clamped S (top), M (middle) and L (bottom) cone to a short section of Gaussian-modulated light input (gray). The background to which modulation was added produced on average 2500 R*/s in each case. Responses were low-pass filtered digitally with a 100 Hz cutoff. Data in this figure is collected from at least 8 retinas. (B) Normalized linear filters of L, M and S cones on a background of 2500 R*/s. Filters were calculated to be those that optimally map Gaussian-noise stimuli to measured cone responses. Thick traces show average filters, thin traces filters from individual cones (17 S cones, 17 M cones, 24 L cones from nine retinas). (C) Comparison of times to peak of S-cone versus LM-cone linear filters. Each open circle represents the times to peak of filters from an S cone and an L or M cone from the same piece of retina (12 S cones from eight retinas). Closed circle plots mean ±sem. S-cone times to peak were significantly shorter than in LM cones (31.5 ± 2 ms for S cones vs 25.2 ± 1.2 ms for paired LM cones, p<0.005, paired t-test). There are fewer cones here than in B because not every S cone could be paired with an LM cone. (D) Comparison of times to peak of M-cone versus L-cone linear filters as in C. L-and M-cone times to peak were not significantly different (p>0.05, paired t-test; 16 M cones across nine retinas, 24.7 ± 0.7 ms for M cones vs 23.8 ± 0.5 ms for paired L cones).
DOI: https://doi.org/10.7554/eLife.39166.007

and LM cones in the same piece of retina (*Figure 4C*; mean ± sem time to peak of 31.5 ± 2 ms for 12 S cones vs 25.2 ± 1.2 ms for paired LM cones). Filters for L and M cones did not differ systematically (*Figure 4D*; mean ± sem time to peak of 24.7 ± 0.7 ms for 16 M cones vs 23.8 ± 0.5 ms for paired L cones).

The results illustrated in *Figures 1–4* indicate that primate S cones have slower responses than LM cones across a range of experimental protocols and retinal regions. The kinetic differences in the cone photovoltages (8.5 ms difference in mean time to peak of matched cones at 5000 R*/cone/s, *Figure 2E*) and photocurrents (6.3 ms difference, *Figure 4C*) were similar, indicating that these differences originate at least largely from differences in phototransduction. Such differences in kinetics could be further enhanced by inner segment conductances (see *Howlett et al., 2017*).

## Adaptation of cone response kinetics differs between S cones and LM cones

Cone photoreceptors operate over a wide range of light levels. Across this range, the kinetics of L- and M-cone flash responses change as they adapt to changing inputs (*Dunn et al., 2007*; *Angueyra and Rieke, 2013*; *Cao et al., 2014*). As shown in *Figure 1B*, the difference in the kinetics between S and LM cones is minimal at low background-light levels and increases with increasing background. Below we explore this background dependence in more detail.

*Figure 5* compares the kinetics of cone light responses across a range of background-light levels. The time to peak of S-cone responses differed minimally across a 50-fold range of background-light levels (*Figure 5A*, data replotted from *Figure 1*). Across the same range of backgrounds, LM-cone responses accelerated substantially as the background increased (*Figure 5B and C*). To control for cell to cell variability, we normalized each cone's times to peak by the time to peak at 5000 R*/s. These normalized response times to peak show little or no dependence on background light level for S cones (<10% change between 1000 and 50,000 R*/s, 95% CI −3% to 12%) and a robust dependence for LM cones (~30% change, L cones 95% CI 26% to 30%, M cones 26% to 30%) (*Figure 5D–F*). Despite these differences in adaptation of response kinetics, adaptation of the response amplitudes did not differ appreciably across cone types (*Figure 5—figure supplement 1*).

We also compared response kinetics by measuring temporal frequency tuning curves at 5000 R*/s and 50,000 R*/s for each cone type (*Figure 5G–I*, data re-plotted from *Figure 3*). As expected based on the flash-response data, the frequency at which S cone sensitivity fell ten-fold changed minimally between 5000 R*/s and 50,000 R*/s (−3.7 to +2.2 Hz, 95% CI). In both L and M cones, there was a significant increase in this frequency at 50,000 R*/s compared to 5000 R*/s (an increase of 3.4 to 11.9 Hz in M cones and 1.7 to 11.5 Hz in L cones, 95% CI).

The experiments of *Figure 5* show that S-cone kinetics vary minimally with background-light level across a range of stimuli. The constancy of their kinetics compared to LM-cone kinetics is interesting mechanistically (see Discussion) and provides a useful difference in signaling properties that can be used to test for contributions of different cone types to downstream cells and to perception (see below and Discussion).

## Differences in cone response properties shape the retinal output

Are the differences in the kinetics and adaptation of S-cone signals maintained throughout retinal circuits and hence could they contribute to visual perception? To answer this question, we compared S- and LM-mediated responses of retinal ganglion cells. A direct comparison of the kinetics of ganglion cell responses mediated by signals originating in different cone types is confounded by the potential for kinetic changes introduced within the circuits conveying the cone signals. Specifically, circuits conveying S-cone signals to ganglion cells could introduce different kinetic changes than circuits conveying LM-cone signals. To mitigate this issue, we relied on the greater sensitivity of the kinetics of LM-cone responses vs S-cone responses to changes in background-light level to test the impact of cone kinetic differences on ganglion-cell responses (*Figure 6*). If differences in cone response kinetics are indeed propagated through the retinal circuitry, the kinetics of S-cone-mediated responses in ganglion cells should change less with background than those of LM-cone-mediated responses, as is the case for the kinetics of the cone responses themselves (*Figure 5*). We tested this prediction by recording S- and LM-cone-mediated responses in small bistratified cells (SBCs).

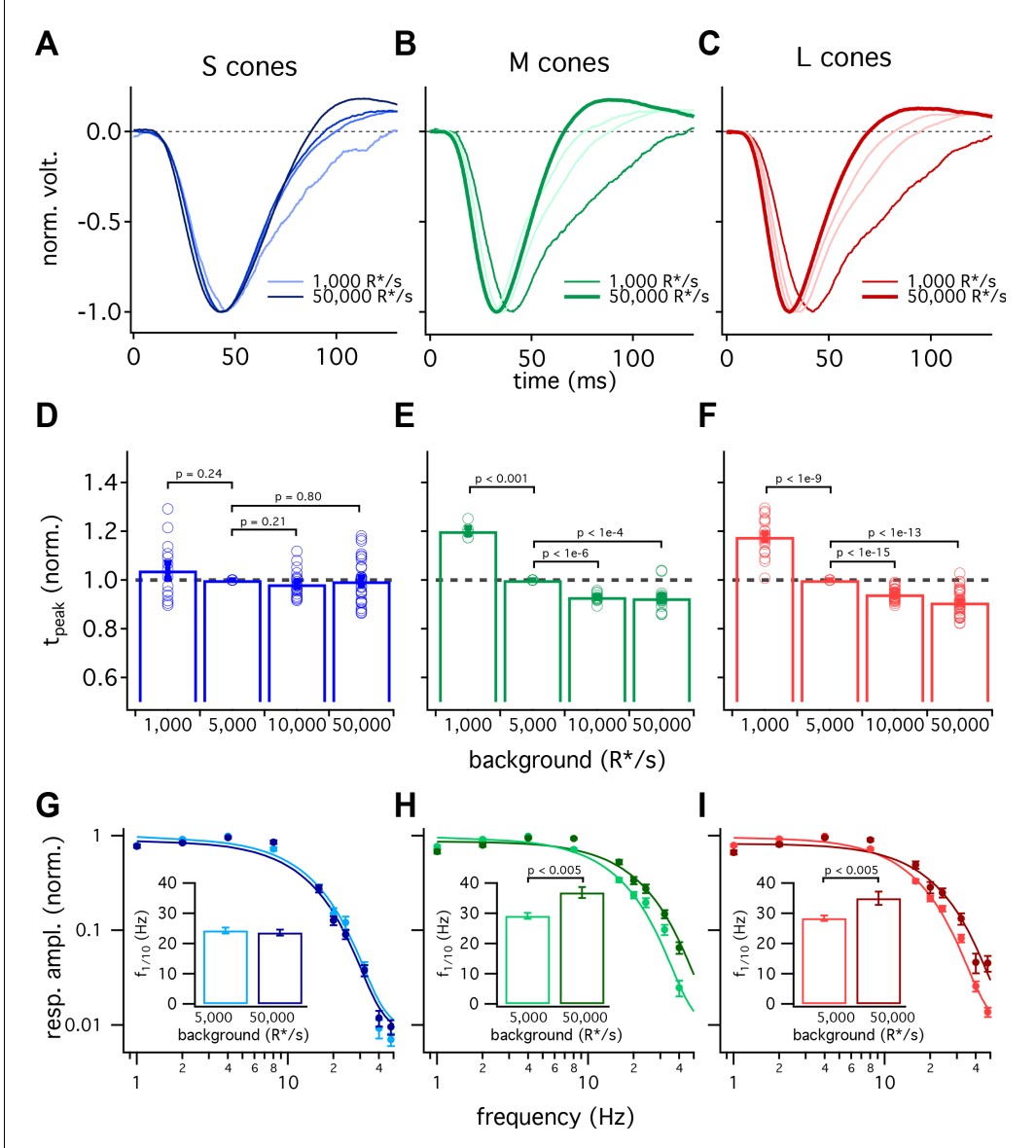

**Figure 5.** S-cone kinetics change minimally across background light levels. (**A–C**) Average normalized voltage responses of S (**A**), M (**B**), and L (**C**) cones on backgrounds of 1000 R*/s (thin line) and 50,000 R*/s (thick line); data replotted from *Figure 1C*. Lighter lines show responses at intermediate light levels of 5000 R*/s and 10,000 R*/s. Total of 36 S cones, 26 M cones and 49 L cones. Data in this figure is collected from >10 retinas. (**D–F**) Mean ± sem relative times to peak across backgrounds in S (**D**), M (**E**), and L (**F**) cones. In each cell, the time to peak at each background was normalized by the time to peak at 5000 R*/s in that cell. p-Values from one-sample t-test. (**G–I**) Frequency-tuning curves at 5000 R*/s and 50,000 R*/s in S (**G**), M (**H**), and L (**I**) cones. Points with error bars are mean ± sem normalized response amplitudes across frequencies. Curves show fit of power spectrum of *Equation 1* to population data. Inset shows mean ± sem frequencies at which response amplitudes decreased to 10% of their maximum value at either background. At 5000 and 50,000 R*/s, these frequencies were 24.4 ± 1.0 Hz (n = 15) and 23.6 ± 1.0 Hz (n = 16) in S cones, 29.2 ± 1.0 Hz (n = 17) and 36.9 ± 1.8 Hz (n = 21) in M cones, and 28.4 ± 0.9 Hz (n = 26) and 35.0 ± 2.2 Hz (n = 17) in L cones. p-Values from unpaired t-test.

DOI: https://doi.org/10.7554/eLife.39166.008

The following figure supplement is available for figure 5:

**Figure supplement 1.** Response amplitude adaptation across cone types.

DOI: https://doi.org/10.7554/eLife.39166.009

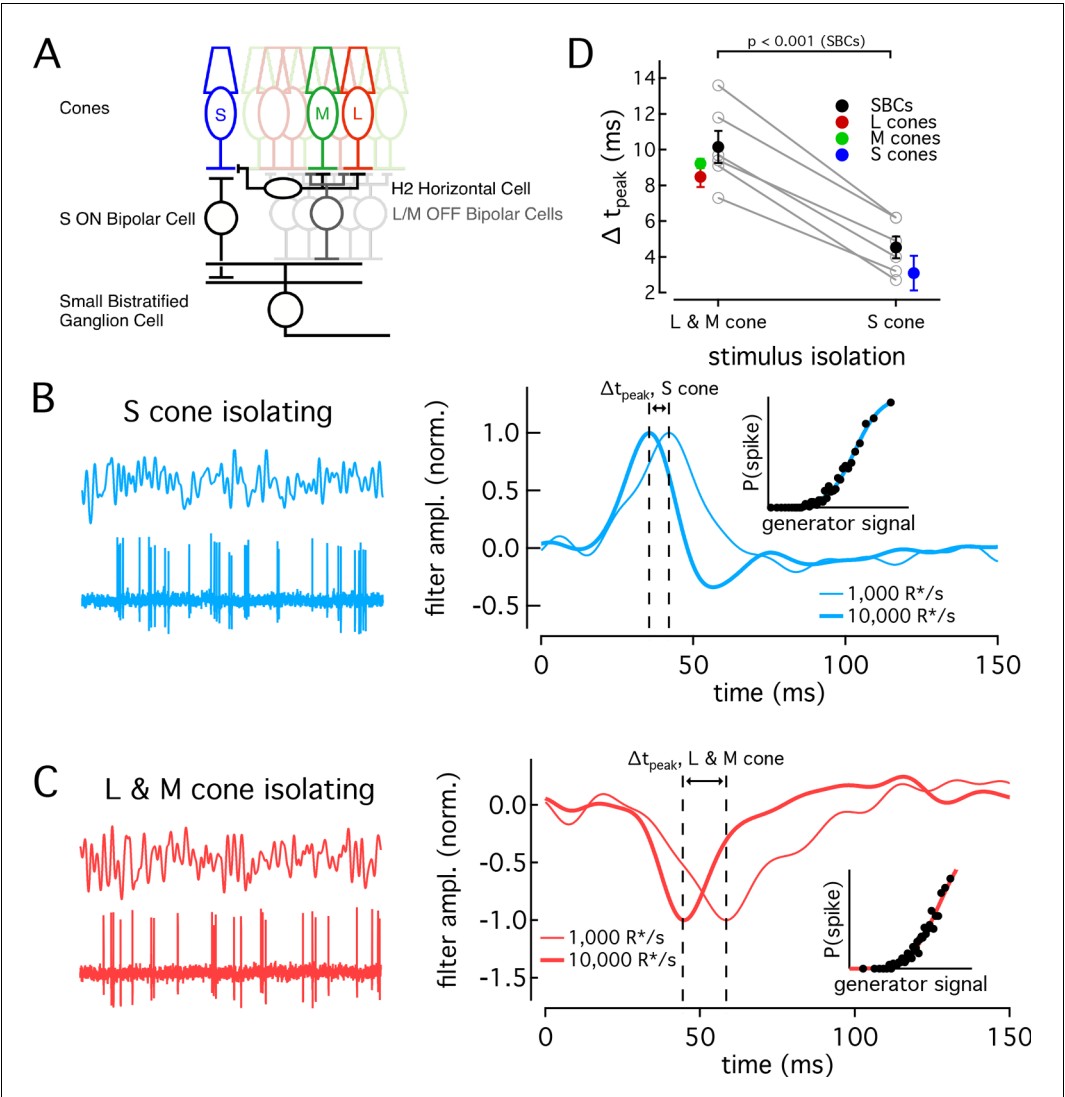

**Figure 6.** Differences in cone adaptation affect retinal ganglion cell responses. (**A**) Previously described circuitry upstream of SBCs. S-ON signals travel from S cones through S-ON Bipolar Cells to SBCs. L/M-OFF signals travel from L and M cones through H2 horizontal cells to S-cone terminals, where they are transmitted to SBCs via S-ON Bipolar cells. Alternatively, L- and M-cone signals may be carried to SBCs directly via an L/M-OFF Bipolar cell. Data in this figure is collected from two retinas. (**B**) Example response of SBC to S-cone isolating Gaussian-noise stimulus (left). Example Linear-nonlinear (LN) model derived from SBC responses to Gaussian-noise stimuli (right). Normalized linear filters shown for backgrounds of 1000 and 10,000 R*/s. Time-to-peak shifts between 1000 R*/s and 10,000 R*s were taken to be the difference in the times to peak of the linear filters at each background. Inset shows nonlinearity mapping from generator signal to spiking probability. (**C**) Example response and LN Model as in (**C**), but for L/M-OFF response. (**D**) Comparison of time-to-peak shifts in S-ON and L/M-OFF SBC responses. Mean ± sem shifts were 10.2 ± 0.9 ms for L/M-OFF responses and 4.5 ± 0.6 ms for S-ON responses (n = 6). Shifts in each cone type between 1,000 and 10,000 R*/s are shown for comparison. Shifts were 3.1 ± 1.0 ms in S cones (n = 14), 9.2 ± 0.3 ms in M cones (n = 7), and 8.49 ± 0.6 ms in L cones (n = 22). P value from paired t-test.
DOI: https://doi.org/10.7554/eLife.39166.010

SBCs stratify in the inner plexiform layer and receive S-cone input via S-ON bipolar cells (*Figure 6A*, *Dacey and Lee, 1994*; *Calkins et al., 1998*). They receive opposite polarity input from L and M cones from either or both of two pathways: (1) via an LM-OFF bipolar cell receiving direct LM-cone input; or (2) via the S-ON bipolar cell which receives indirect LM input that reaches S cones through the H2 horizontal cell (*Field et al., 2007*; *Crook et al., 2009*; *Packer et al., 2010*). As discussed above, these differences in the route that S- and LM-cone signals take to reach the SBC precluded directly comparing the kinetics of their S-ON and LM-OFF responses. Nonetheless, if cone kinetics influence retinal output, we predict that the kinetics of the S-ON response in SBCs would

change significantly less across light levels than the kinetics of the LM-OFF response. To test this, we compared changes in the kinetics of the S-ON versus LM-OFF responses.

We recorded SBC spike responses to S-cone-isolating or LM-cone-isolating Gaussian noise stimuli (*Figure 6B and C*; see Materials and methods for stimulus construction). From these data, we constructed a linear-nonlinear model (LN model) consisting of the combination of a linear filter and a static nonlinearity that best describe the mapping from stimuli to spikes. The linear filter in this model captures the response kinetics while the static nonlinearity accounts for a nonlinear dependence of the response on the stimulus. The kinetics of the SBC responses were quantified as the times to peak of the linear filters (see Materials and methods).

As shown in the right-hand panels of *Figure 6B and C*, the linear filters computed for S-cone-isolating and LM-cone-isolating stimuli exhibit the expected polarity for a cell generating S-ON and LM-OFF responses. As observed previously (*Field et al., 2007*), S-ON responses at these low-light levels are faster than LM-OFF responses, consistent with the extra synapse in the circuits conveying LM- vs S-cone signals to SBCs. At 1000 R*/cone/s, S cone filters reached a peak in 38.0 ms, while LM cone filters peaked in 54.9 ms. At 10,000 R*/s, times to peak were 33.4 ms for S cones and 44.7 ms for LM cones. These responses are somewhat faster than those reported in *Field et al. (2007)*, but the difference in kinetics between S and LM responses at the lower light level is similar.

The kinetics of the responses became faster between a background of 1000 R*/s and 10,000 R*/s for both S-ON and LM-OFF stimuli. For each cell, we calculated the time-to-peak shift between 1000 R*/s and 10,000 R*/s for the linear filters from the S-isolating and LM-isolating stimuli (*Figure 6D*). In each recorded cell, the shift in time to peak of the S-cone mediated responses was smaller than that of the LM-cone mediated responses, as predicted from the cone data (*Figure 6D*). Furthermore, across the population, the average time-to-peak shift for S-cone mediated responses was significantly less than that for LM-cone mediated responses (S-ON shift 4.5 ± 0.6 ms and LM-OFF shift 10.2 ± 0.9 ms, mean ±sem from 6 SBCs). The average shifts seen in the population of SBCs agreed well with the average shifts seen in the cones themselves (*Figure 6D*).

The SBC recordings indicate that differences in kinetics of cone responses shape retinal output signals. Further, they highlight that differences in the background dependence of S- vs LM-cone kinetics provide a tool to probe the impact of different cone types on downstream signaling.

## Noise in S-cones differs from that in LM-cones

Signal and noise together determine what information cones transmit to downstream circuitry and what limitations cone responses impose on visual perception. Noise arises at multiple stages in the biochemical cascade of phototransduction (*Schneeweis and Schnapf, 1999*; *Angueyra and Rieke, 2013*). One source is thermal activation of the photoreceptor photopigment, and pigment thermal stability has been shown to depend on the wavelength of peak sensitivity (*Barlow, 1957*; *Luo et al., 2011*). This hypothesis predicts that photoreceptors sensitive to longer wavelengths will be noisier; this prediction holds in salamander cones (*Rieke and Baylor, 2000*). Noise in primate cones appears to originate primarily from sources other than photopigment thermal activation. Thus, it is unclear how noise will vary across cone types (*Schneeweis and Schnapf, 1999*; *Angueyra and Rieke, 2013*).

We recorded cellular noise across a range of background light levels in S, M, and L cones (*Figure 7A and B*, see Materials and methods and Angueyra and Rieke, 2013) and compared the measured noise to dim-flash responses at each background (*Figure 7C*). Noise extended to frequencies far higher than the flash-response spectrum (*Figure 7C*, *Angueyra and Rieke, 2013*). Hence, we summarized our noise measurements by integrating noise across two frequency ranges – one that overlaps with the flash response and one that does not. We refer to these as the flash-response range and high-frequency ranges (*Figure 7C*).

As previously seen in salamander and primate LM cones (*Rieke and Baylor, 2000*; *Angueyra and Rieke, 2013*), noise changes as a function of background-light level (*Figure 7A*). Previous measurements from primate LM cones showed that as light levels increased from darkness, noise in the flash-response range increased ~2 fold and then fell sharply (*Angueyra and Rieke, 2013*). This was attributed to the background light introducing Poisson noise in photon absorption while being too dim to engage adaptation. At higher backgrounds, response adaptation appeared to outweigh the increased noise in photon absorption, leading to decreased noise in the flash-response range. L, M and S cones all followed this behavior, but the increase in noise in S cones was significantly smaller

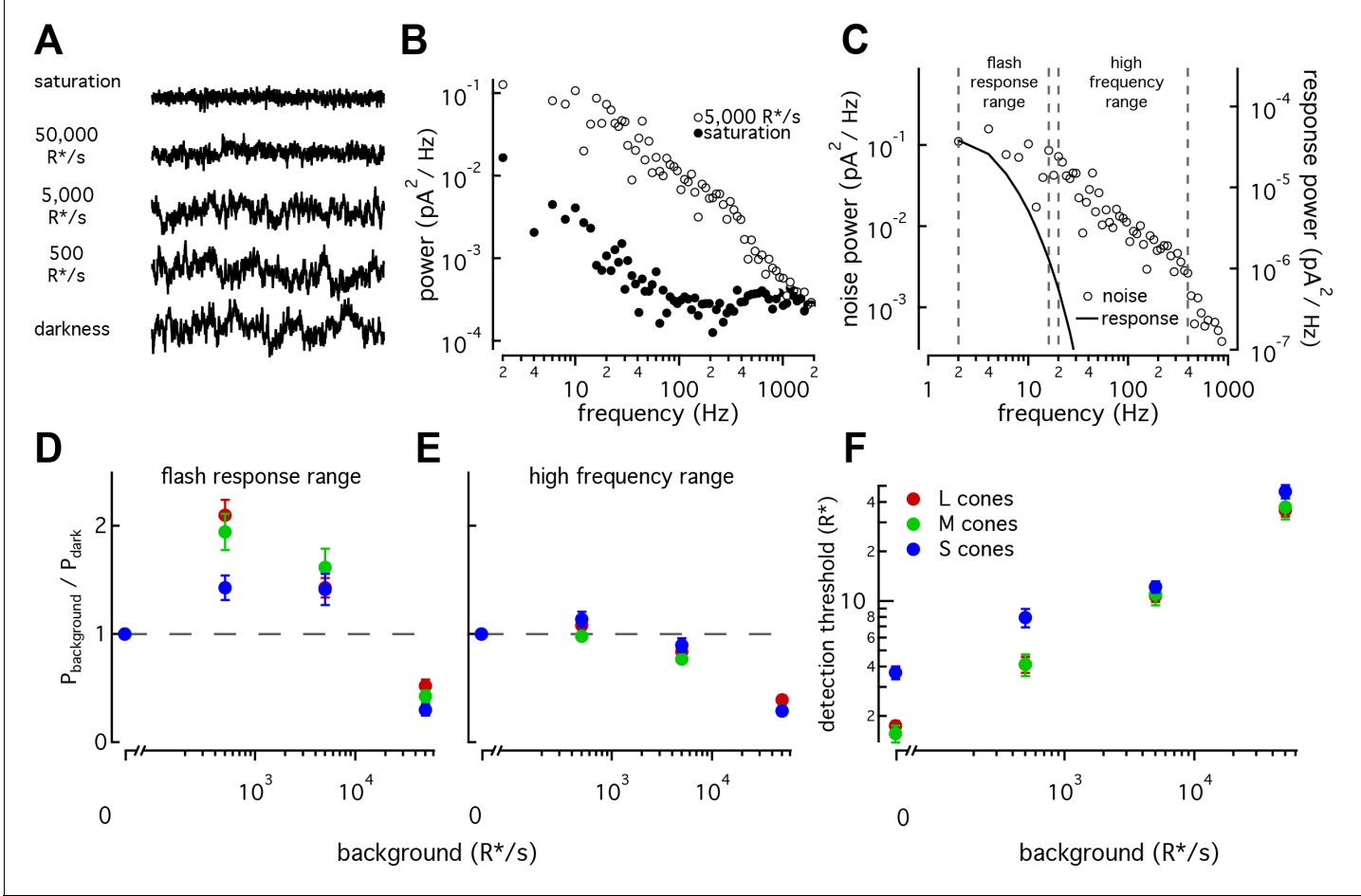

**Figure 7.** S cones have lower signal-to-noise ratios under dim-lighting conditions. (**A**) Example current recordings from an S cone on backgrounds of 0, 500, 5000, and 50,000 R*/s and in saturating light. Data in this figure is collected from >10 retinas. (**B**) Example power spectra of noise at 5000 R*/s (open circles) and in saturating light (closed circles) from an S cone. Spectrum at 5000 R*/s includes contributions from cellular as well as instrumental noise. (**C**) Example isolated cellular noise spectrum (left axis, open circles) and flash response spectrum (right axis, solid line), both on a background of 5000 R*/s. Dashed vertical lines identify bounds for flash-response and high-frequency ranges used in (**D–F**) (Flash-response range: 2–16 Hz; high-frequency range: 20–394 Hz). (**D**) Noise power in flash-response range in S (n = 14), M (n = 18), and L (n = 29) cones. Values show mean ± sem powers across backgrounds normalized by the power in darkness. Dashed horizontal line shows noise power in darkness. The change in S-cone noise at 500 R*/s was significantly lower than that in L and M cones (S vs M, p<0.05; S vs L, p<0.001; unpaired t-test). (**E**) As in (**D**) but for noise in high-frequency range. (**F**) Detection thresholds across backgrounds in S (n = 9), M (n = 9) and L (n = 15) cones. Plotted values show mean ±sem. In darkness, mean ± sem threshold is 3.67 ± 0.33 R* in S cones, 1.57 ± 0.18 R* in M cones, and 1.75 ± 0.12 R* in L cones. At 500 R*/s it is 7.95 ± 1.02 R* in S cones, 4.13 ± 0.63 R* in M cones, and 4.12 ± 0.46 R* in L cones.

DOI: https://doi.org/10.7554/eLife.39166.011

The following figure supplement is available for figure 7:

**Figure supplement 1.** Noise-effective isomerizations across cone types Effective noise-isomerization levels across backgrounds.

DOI: https://doi.org/10.7554/eLife.39166.012

than that in LM cones at 500 R*/s (*Figure 7D*; S-cone noise increases by factor of 1.4 ± 0.1, M-cone noise by 1.94 ± 0.17, and L-cone noise by 2.10 ± 0.14, mean ±sem). Such a difference could arise if the S-cone response to a single-photon absorption was smaller relative to noise compared to the LM-cone single photon response; a smaller single photon response would in turn mean less noise due to Poisson fluctuations and less of a dependence of noise on background. Two predictions can be made if a smaller single-photon response is the main source of differences in the noise properties across cone types: (1) adaptation of S-cone noise in the high-frequency range, where noise in photon absorption should have little effect, should be similar to that in LM cones; and (2) the number of

isomerizations required to match the noise power should be higher in S-cones at these backgrounds. Both predictions held true (*Figure 7E*; *Figure 7—figure supplement 1*).

The increased noise relative to the flash response in S cones at 0 and 500 R*/s suggests that detection thresholds may also be higher in S cones. Although noise-effective isomerizations and detection thresholds are closely linked, we calculated brief-flash detection thresholds in each cone type to facilitate comparison with previous work. The detection threshold was defined as the flash-response strength needed to match the noise power in a 200 ms integration window (*Figure 7F*). As expected, S-cone detection thresholds were higher than those of LM cones both in darkness (3.7 ± 0.3 R* in 9 S cones vs 1.7 ± 0.1 R* in 24 L and M cones) and at 500 R*/s (8 ± 1 R* for S cones vs 4.1 ± 0.5 R* for LM cones). Thresholds were similar across cone types at 5000 and 50,000 R*/s.

The results described here show that, relative to their respective flash responses, S-cone noise is higher than LM-cone noise at low-light levels and comparable at higher-light levels. This is contrary to predictions if noise was dominated by thermal activation of the photopigment and the thermal-activation rate scaled with wavelength of peak sensitivity. The difference in cone noise is sufficiently large to impact cone detection thresholds and other perceptual tasks based on the cone signals (see Discussion).

## Discussion

Visual sensitivity to time-varying signals originating in S vs LM cones differs, and this difference has been suggested to originate in the cones themselves (*Lee et al., 2009*). Due to limited intracellular recordings from primate cones, this suggestion has not been well tested. Motivated by studies of cones across spectral types in other species (*Rieke and Baylor, 2000*; *Howlett et al., 2017*) as well as recently reported differences in primate cones across eccentricity (*Sinha et al., 2017*), we aimed to test this suggestion. We find that S-cone responses differ from those of LM cones in three ways. First, S-cone responses are slower than LM-cone responses in both peripheral and foveal retina. Second, the kinetics of S-cone responses adapt minimally across background light levels, unlike responses of LM cones. Third, S cones have a lower signal-to-noise ratio at low-light levels compared to LM cones, resulting in higher S-cone flash-detection thresholds. These differences shape the output of the retina and are predicted to affect perception.

### Different kinetics of signals in S- vs LM-cone pathways begin at the first neuron of the visual system

S cones produce slower flash responses and respond less robustly to high-frequency sinusoidal stimuli than LM cones. The similarity of these differences in the cone current and voltage responses indicates that they originate at least in large part within the phototransduction cascade. These differences likely contribute to our lower perceptual sensitivity to rapidly modulated signals that preferentially stimulate S cones compared to those that preferentially stimulate LM cones (*Brindley et al., 1966*; *Marks and Bornstein, 1973*; *Smithson and Mollon, 2004*). Furthermore, the kinetics of S-cone responses changed minimally with increasing light level, while responses of LM cones sped considerably (*Figure 5*). Perceptual experiments find a similar difference in the dependence of flicker-fusion frequencies on background-light level for chromatic stimuli encoded by S vs LM cones (*Brindley et al., 1966*; *Marks and Bornstein, 1973*); specifically, over a sizable range of light levels, the flicker-fusion frequency remains near constant at short wavelengths while increasing at long wavelengths. Further, perceptual chromatic-discrimination experiments indicate that S-cone responses are effectively delayed by ~12 ms relative to opposing LM-cone responses (*Lee et al., 2009*); this delay applies to discrimination in many different color directions, consistent with an origin prior to the combination of cone signals into cone-opponent pathways. This perceptual delay compares closely with the 8–10 ms difference in the time to peak of S- vs LM-cone responses that we observe (*Figure 2*).

The relevance of the kinetic differences across cones types for retinal output signals is supported by the differences we found in the response kinetics of SBCs (*Figure 6*). Previous measurements of SBC-response kinetics at similar background-light levels found that LM-cone responses are slower than S-cone responses (*Field et al., 2007*); this is consistent with the extra synapse in the circuitry conveying LM-cone signals to SBCs compared to that conveying S-cone signals. These differences in circuitry, however, mean that absolute response kinetics may not reflect the cone kinetics. We relied

instead on the dependence of the response kinetics on background-light level. We found that the kinetics of SBC responses to S-cone-isolating stimuli changed less with increasing backgrounds than responses to LM-cone-isolating stimuli. Thus, differences in response kinetics across cone types appear to be propagated through the retinal circuitry subserving SBCs. It remains to be seen whether these differences are preserved in the other parallel pathways carrying S-cone signals. This approach also demonstrates that differences in the background dependence of response kinetics across cone types can be used to probe the contributions of different cone types to downstream circuitry and to perception.

S cones also contribute to luminance perception, and these contributions differ in several ways from the S-cone contributions to color perception (*Lee and Stromeyer, 1989*; *Stockman et al., 1991*). Notably, S-cone contributions to high-frequency luminance flicker increase in strength with increasing light level, indicating a speeding of the kinetics of the underlying neural responses (*Stockman et al., 2007*). This speeding is similar to that observed for M-cone contributions to luminance perception (*Stockman et al., 2006*) as well as that observed in L- and M-cone inputs to H1 horizontal cells (*Smith et al., 2001*). The change in kinetics observed perceptually and in horizontal cells, however, exceeds the change we measure in the cone responses. Together these observations suggest that processes downstream of the cone photovoltage substantially shape cone responses, and that these circuit mechanisms differentially shape S-cone responses in chromatic and luminance pathways. Perception is also shaped by interactions between signals initiated in different cone types - for example perceptual sensitivity to S-cone-mediated signals is reduced by activity of L and M cones (*Wisowaty and Boynton, 1980*). It will be interesting to determine the contributions of retinal circuits to these additional processes shaping S-cone-mediated responses.

In addition to comparing S cones to LM cones, we directly compared a large dataset of L and M cones. Their responses were identical within the resolution of our measurements. Recent work similarly shows minimal differences in gene expression between L and M cones (*Peng et al., 2018*). Given that S cones diverged evolutionarily from the common ancestor of LM cones far earlier than L and M cones diverged from each other, it is not surprising that S cones differ more from LM cones than L and M cones differ from each other (*Nathans et al., 1986*). Differences in photopigment stability, however, could cause L and M cones to differ in adaptational state and hence in kinetics. Our results tightly constrain any such differences.

## Noise differs across cone spectral types

The origin of noise in cone photoreceptors differs between salamander and primate. Specifically, isomerization-like events in salamander L cones contribute significantly to their noise, while noise in primate LM cones is dominated by other sources (*Schneeweis and Schnapf, 1999*; *Rieke and Baylor, 2000*; *Angueyra and Rieke, 2013*). Compared to L cones, salamander S cones have much lower noise, consistent with a more stable photopigment. This is consistent with the expected wavelength dependence of photopigment stability (*Luo et al., 2011*). Unlike the situation in salamander, the effective noise of primate S cones exceeds that of LM cones at low background-light levels (*Figure 7*). This results in higher flash-detection thresholds for S vs LM cones.

## Possible mechanistic origins of differences in cone signals

S- vs LM-cone photocurrents and voltages exhibited similar differences in kinetics (*Figures 1–4*), suggesting an origin largely from differences in the cone phototransduction machinery. This is similar to salamander S vs L cones (*Rieke and Baylor, 2000*), but different from goldfish S vs LM cones, in which kinetic differences originate from the expression of a hyperpolarization-activated cyclic nucleotide-gated (HCN) channel in LM but not S cones (*Howlett et al., 2017*). Mammalian rod photoreceptors evolved from S cones and the two photoreceptor types share several phototransduction components (*Craft et al., 2014*). For instance, primate rods and S cones share the same arrestin isoform (arrestin1) responsible for photopigment inactivation and hence recovery of the light response. S cones also express the cone arrestin isoform (arrestin3), but the expression is lower than that in LM cones (*Craft et al., 2014*). Based on these and other biochemical differences, S cones are thought to share functional features with both rods and LM cones.

Both the rising and recovery phases differ between S- and LM-cone responses (*Figure 2—figure supplement 2*). These two phases of the light responses are primarily controlled by different aspects

of the phototransduction cascade. Hence, this analysis suggests that multiple components of the phototransduction cascade differ between S and LM cones, much like the differences observed between mammalian rods and cones (*Ingram et al., 2016*).

Responses of both rods and cones typically speed with increasing background-light level. This speeding can largely be explained by the increase in the rate of cGMP turnover produced by background light (*Nikonov et al., 2000*). The constancy of the kinetics of the S-cone responses with increasing background (*Figure 5*) suggests that either the cGMP-turnover rate contributes minimally to response kinetics, or one or more components of the phototransduction cascade must slow with background to compensate the expected increase in cGMP-turnover rate.

### Implications of cone differences in interpreting visual system function

Considerable effort has gone into understanding retinal circuits due to their importance in explaining human vision as well as their experimental accessibility. Such work often aims to understand and ultimately model the transformation these circuits perform between light arrival and some measured response. These models attempt to assign different computations to different circuit elements. Cones perform a complex transformation on their inputs (*Endeman and Kamermans, 2010*), thus making it difficult to differentiate their contributions from those of downstream circuitry to the computations that support vision (*van Hateren, 2005*). One approach to this issue is incorporating reliable predictions of cone responses into retinal models in an attempt to better define the inputs to downstream circuitry. Only recently has it become possible to use validated models of the differences in LM cones across the visual field (*Sinha et al., 2017*). Our results here expand upon this by providing the information necessary to construct models that capture differences between different spectral types of primate cones.

## Materials and methods

### Key resources table

| Reagent type (species or resource | Designation | Source or reference | Identifiers | Additional information |
|---|---|---|---|---|
| Antibody | anti-OPN1SW (Mouse monoclonal) | Santa CruzBiotechnology | Cat#: sc14363 | (1:50) |
| Antibody | anti-goat IgG (H + L) HiLyte Fluor 750 | AnaSpec | | (1:100) Secondary antibody |
| Biological Samples | Macaque retina | Washington Regional Primate Research Centre | N/A | *Macaca fascicularis*, *Macaca nemestrina*, and *Macaca mulatta* of both sexes, aged 2 through 20years |
| Chemical compound, drug | Ames | Sigma | 1420 | |
| Chemical compound, drug | DNase1 | Sigma | 11284932001 | |
| Software, Algorithms | IGOR Pro | WaveMetrics | https://www.wavemetrics.com/ | |
| Software, Algorithms | MATLAB | Mathworks | https://ch.mathworks.com/products/matlab | |
| Software, Algorithms | Symphony | Symphony-DAS | http://symphony-das.github.io/ | |

## Tissue, cells, and solutions

Electrophysiological recordings were performed on primate retina obtained through the Tissue Distribution Program of the University of Washington's Regional Primate Research Center. Recordings were made from retinas from *Macaca fascicularis*, *Macaca nemestrina*, and *Macaca mulatta* of both sexes, aged 2 through 20 years. All use of primate tissue was in accordance with the University of Washington Institutional Animal Care and Use Committee. Tissue was obtained and prepared as described previously (*Angueyra and Rieke, 2013*; *Sinha et al., 2017*). In short, retina was dark adapted (>1 hr) and stored in warm (32° C), oxygenated Ames medium; this time was sufficient to fully dark-adapt the retina, as judged by responses to single photons in rods and downstream cells (*Ahnelt et al., 1987*; *Ala-Laurila and Rieke, 2014*). Following dark adaptation, a piece of retina roughly 2–3 mm on a side was separated from the pigment epithelium and placed photoreceptor side up (cone recordings) or down (retinal ganglion-cell recordings) on a poly-lysine-coated coverslip (BD Biosciences, San Jose, CA) that served as the floor of our recording chamber. For cone recordings, retina was treated with DNase1 (Sigma-Aldrich, St. Louis, MO) (30 units in ~1 mL for ~3 min) prior to placing it in the recording chamber. Throughout recordings, the tissue was continuously perfused with warm, oxygenated Ames solution.

## Patch-clamp recordings

Cone patch-clamp recordings were performed in intact pieces of flat-mounted retina as described previously (*Angueyra and Rieke, 2013*; *Sinha et al., 2017*). In short, we measured cone light responses using a combination of whole-cell voltage-clamp (holding potential = −60 mV; not corrected for liquid junction potential) and current-clamp (holding current = 0 pA) recordings. Extracellular recordings from retinal ganglion cells were performed as described previously (*Sinha et al., 2017*). Data were low pass-filtered at 3 kHz, digitized at 10 kHz, and acquired using a Multiclamp 700B amplifier. No additional filtering was applied to any of the data presented except in *Figure 4A*. All recordings were controlled using Symphony Data Acquisition Software, an opensource, MATLAB-based electrophysiology software (https://github.com/symphony-das).

## S cone identification

S cones make up a minority of the cone photoreceptors within the primate retina. While recording, the photoreceptor array was visualized using DIC microscopy, making all cones appear similar. Initially, we attempted to label S cones in in vitro retina using an antibody directed against the S-opsin molecule (anti-OPN1SW, sc-14363, Santa Cruz Biotechnology, Dallas, TX). Although we did not use this approach to collect any of the data reported here, it did help us learn to identify S cones based on their morphology and position within the photoreceptor array (*Ahnelt et al., 1987*; *Packer et al., 2010*). Targeting cones that appeared slightly smaller, recessed, and out of place within the photoreceptor array dramatically increased the probability that the cones were S cones. Targeting cones in this manner allowed us to efficiently collect a large number of S-cone recordings.

## Cell-selection criteria

We selected cells for data collection based on the amplitude of their response to a bright flash. For current-clamp recordings, data was collected only from cones with a maximal response exceeding 10 mV. For voltage-clamp recordings, this criterion was 100 pA (at a holding voltage of −60 mV). The assumption underlying these criteria is that the cells with the largest responses most closely resemble cells in vivo. We controlled for three additional factors that could potentially bias our results. First, cone responses slow over time during whole-cell recordings, likely due to washout of intracellular components essential for phototransduction. We collected data for no longer than 4 min after initiating a recording to minimize this source of bias; time to peak and response amplitude changed minimally during this time, and the response waveform did not noticeably change. Second, the data reported here was collected between 1 and 15 hr after collecting the retina, and cone responses could change over this time. To check this possibility, we grouped the cells into periods of 1–6, 6–10, and 10–15 hr post-retina collection. The time-to-peak of S-cone responses was significantly different from that of LM cones in each time window (data not shown). Third, responses can differ from one retina to the next. To control for such retina-to-retina differences, we referenced each recorded S cone to L and/or M cones recorded in the same retina. The differences we observe

in responses collected across all recorded cones (*Figure 1*) were also present in responses of cells within a given piece of retina (*Figure 2E and F* and *Figure 4C*). The data reported here represents recordings from a total of 72 S cones, 60 M cones and 112 L cones from peripheral retina and 32 S cones, 25 M cones, and 28 L cones from foveal retina. For each dataset except *Figure 6*, we used at least five retinas from five animals. In *Figure 6*, we used two retinas from two animals.

## Light stimulation

Stimuli were presented from computer-driven LEDs with peak wavelengths of 406, 515, and 640 nm to provide the flexibility to effectively stimulate all three cone types. Light stimuli covered a ~ 500 μm disk centered on the targeted cell. Following a successful recording from peripheral retina, we moved to a location on the retina outside the region exposed to light before attempting another recording; this ensured that all recorded cells were fully dark adapted at the start of the recording. This was limited to 2–3 locations per fovea given its small size. The minimum light level used (500 R*/cone/s) was sufficient to effectively eliminate rod responses (*Grimes et al., 2018*).

All stimulus protocols were generated using custom-written MATLAB-based extensions of Symphony Data Acquisition Software, and delivered at 10 kHz. To calculate cone isomerization rates we measured LED spectra, used LED-power measurements, primate photoreceptor spectra from *Baylor et al. (1987)*, and an effective collecting area of 0.37 μm$^2$ (*Schnapf et al., 1990*). For reference, one photopic troland is 10–30 R*/cone/s (*Spillmann and Werner, 1990*; *Crook et al., 2009*). Based on morphological differences, S cones could have a smaller collecting area than LM cones (*Ahnelt et al., 1987*; *Packer et al., 2010*); however, such differences are not likely to explain our results, as the responses of S cones at backgrounds of 50,000 R*/cone/s are slower than those of LM cones at 5000 R*/cone/s. Furthermore, cone collecting areas estimated for light delivered from above and below the retina (i.e. directly to the outer segment vs first traversing the inner segment) were similar; thus, focusing of light via the inner segment contributes minimally in our preparations, and hence differences in inner segment size between S and LM cones are unlikely to affect the collecting area.

Cone-isolating stimuli were constructed using a matrix that mapped from LED input to our calculated isomerizations in each cone type. The inverse of this matrix maps from isomerizations in each cone type to an input to each LED. Using this matrix, we were able to specify our stimuli in terms of isomerizations to each cone type. Any failures in isolating S cones versus LM cones would decrease the magnitude of the kinetic differences we saw in the analysis presented in *Figure 6*.

To compute the linear filters in *Figures 4* and *6*, we presented time-varying Gaussian-noise stimuli with a 50% contrast (SD/mean) and 0–60 Hz bandwidth. This stimulus was presented at a mean luminance of 2500 R*/cone/s for the experiments of *Figure 4* and either 1000 or 10,000 R*/cone/sec for the experiments of *Figure 6*.

Noise recordings in *Figure 7* were based on 3 s light steps from darkness to different background-light levels. In a subset of cells, the step duration was increased and dim flashes were superimposed upon the step to provide the data necessary to compute detection thresholds.

## Analysis

All data were analyzed using custom-written MATLAB analysis routines.

### Time-to-peak calculation

All time-to-peak analyses were repeated using a series of different techniques to calculate the times to peak of cone flash responses and small bistratified ganglion-cell linear filters. Results remained significant regardless of which technique was used. For the first approach, we took the time at which the raw average response or linear filter reached its maximal value. Due to unavoidable noise, it was apparent in some recordings that a random spike in noise had affected the time to peak determination. To control for this, we used two fitting-based approaches. For the first, we fit a truncated Gaussian distribution spanning ~20 ms to the peak region of the average flash response or linear filter. The time to peak was taken to be the time at which the Gaussian fit reached its maximal value. The final approach involved fitting a function previously shown (Angueyra and Rieke, 2013) to capture the structure of the flash response (*Equation 1*). We found this function to have the representational power necessary to fit both cone flash responses and SBC linear filters and we defined the

time to peak as the time at which this fit function reached its maximal value. For cones, all times-to-peak response reported here were calculated using the truncated Gaussian fitting technique. Small bistratified cell linear filter times to peak were calculated using fits from *Equation 1*.

$$f(t) = \alpha \times \left[ \frac{\left(\frac{t}{t_{rise}}\right)^4}{1 + \left(\frac{t}{t_{rise}}\right)^4} \right] \times \left[ e^{-\left(\frac{t}{t_{decay}}\right)} \right] \times \left[ cos\left(\frac{2\pi t}{t_{osc}} + \phi\right) \right] \tag{1}$$

## Spectral analysis

Noise and flash-response power spectra were calculated using MATLAB's built-in fast Fourier transform and converted to two-sided power spectral densities with units of $pA^2$/Hz. Dim-flash response recordings contain a combination of cellular noise, instrumental noise, and the flash response. To isolate the flash-response power, power-spectral densities were computed using fits to the dim-flash response (using *Equation 1*). To compute the power in different frequency ranges, the power spectral densities were integrated across the range.

## Noise isolation

Cellular-noise isolation was performed as in *Angueyra and Rieke (2013)*. Any current fluctuations in a voltage-clamp recordings are a combination of noise arising in phototransduction in the cones (cellular noise) and noise from the recording itself (instrumental noise). Providing a near-saturating light stimulus shuts down phototransduction and isolates instrumental noise. Under the assumption that cellular and instrumental noise are independent, cellular noise can be isolated by subtracting the power spectrum of the noise in saturating light from the noise power spectrum at each background.

## Temporal frequency-tuning curves

Frequency-tuning curves were constructed using a cone's responses to sinusoidal stimuli across a range of frequencies. To quantify a cone's response amplitude at a given frequency, the best fit was found using the following equation:

$$y = a * sin(2 * pi * fx + b) + c \tag{2}$$

$f$ was matched to the frequency of the stimulus. $a$, $b$ and $c$ were free to vary. The response amplitude was taken to be the fit value of $a$ divided by the contrast of the stimulus. This contrast normalization step was necessary because higher contrasts were required to elicit responses at higher frequencies where the cells were less responsive. Before averaging tuning curves across cells, the tuning curve of each cell was normalized such that its amplitude at the frequency with the strongest response was 1.

The frequency at which a cone's response decreased by a factor of 10 was calculated by interpolating a smooth function fit to its frequency-response curve. Under an assumption of linearity, the shape of the frequency-tuning curve is equivalent to the power spectrum of the cone flash response. Therefore, the best fit was found for each curve using the power spectrum of *Equation 1*. Best fits were found using the following loss function:

$$L(\theta) = \sum_i \left| log\left(\frac{F(\omega_i, \theta)}{D(\omega_i)}\right) \right| \tag{3}$$

where $F(\omega_i, \theta)$ is the prediction from a fit with parameters $\theta$ at the frequency $\omega_i$ and $D(\omega_i)$ is the data.

*Adaptation Curves:* For each cell, average dim-flash responses across a range of background-light levels were fit with *Equation 1*. The amplitude of such a response was taken to be the amplitude of the fit function and converted to a response per isomerization by dividing by the flash strength. A cell's response amplitudes per isomerization across background light levels were fit with a Weber curve:

$$\frac{\gamma_B}{\gamma_D} = \frac{1}{\left(1 + \frac{I_B}{I_0}\right)} \tag{4}$$

The half-maximum amplitude of the adaptation curve was taken to be the value of $I_0$ from the fit. Fits were performed using the loss function from Equation 3. Prior to averaging adaptation curves

across cells, each was scaled such that its best fit to Equation 4 would have a response per isomerization of 1 on a background of 0 R*/s.

## Cone linear-filter calculation

Linear filters were computed using cone responses to Gaussian-noise stimuli as described previously (*Wiener, 1949*; *Rieke et al., 1997*; *Chichilnisky, 2001*).

## LN-model construction

LN-models were constructed from small bistratified ganglion-cell responses to Gaussian-noise stimuli through a series of steps. First, spike detection was performed. Then, the optimal linear filter mapping from the stimulus to binary vectors of spike responses was computed (*Chichilnisky, 2001*). Finally, a nonlinearity was calculated to map the output of a stimulus convolved with this linear filter (generator signal) to a probability of spiking. This was constructed by convolving each white-noise stimulus vector with the calculated linear filter and, based on the detected spikes, determining the probability of a spike given some generator signal. Nonlinearities were fit with Gaussian cumulative-distribution functions.

## Acknowledgements

We thank Shellee Cunnington and Mark Cafaro for excellent technical support. Retinal tissue was provided by the Washington National Primate Research Center's Tissue Distribution Program. We are grateful for the efforts of WaNPRC staff, especially Chris English and Drew May. Mike Manookin assisted in tissue preparation and provided helpful feedback on experiments. Greg Horwitz provided helpful feedback on an earlier version of this manuscript. Christian Puller suggested using antibody labeling to identify S-cones in live retina. This work was supported by NIH grants R01-EY011850 (FR), R01-EY028111 (FR), R00-EY026070 (RS), the Simons Foundation (FR), the Howard Hughes Medical Institute (FR), and the ARCS Foundation (JB).

## Additional information

### Competing interests

Fred Rieke: Reviewing editor, *eLife*. Jacob Baudin: JB is now employed by Google Inc. All work by JB on this manuscript was completed prior to joining Google Inc. The other authors declare that no competing interests exist.

### Funding

| Funder | Grant reference number | Author |
|---|---|---|
| National Institutes of Health | EY011850 | Fred Rieke |
| Howard Hughes Medical Institute | Investigator | Fred Rieke |
| Simons Foundation | Fellowship | Fred Rieke |
| ARCS Foundation | Scholar | Jacob Baudin |
| National Institutes of Health | EY028111 | Fred Rieke |
| National Institutes of Health | EY026070 | Raunak Sinha |

The funders had no role in study design, data collection and interpretation, or the decision to submit the work for publication.

### Author contributions

Jacob Baudin, Conceptualization, Data curation, Software, Formal analysis, Investigation, Methodology, Writing—original draft, Writing—review and editing; Juan M Angueyra, Conceptualization, Methodology, Writing—review and editing; Raunak Sinha, Conceptualization, Resources, Data

curation, Formal analysis, Supervision, Funding acquisition, Methodology, Writing—original draft, Writing—review and editing; Fred Rieke, Conceptualization, Formal analysis, Supervision, Investigation, Visualization, Methodology, Writing—original draft, Writing—review and editing

### Author ORCIDs
Juan M Angueyra 
Raunak Sinha 
Fred Rieke 

### Decision letter and Author response
Decision letter https://doi.org/10.7554/eLife.39166.019
Author response https://doi.org/10.7554/eLife.39166.020

## Additional files

### Supplementary files
• Transparent reporting form
DOI: https://doi.org/10.7554/eLife.39166.013

### Data availability
Source data are available via Dryad (http://dx.doi.org/10.5061/dryad.gv5k2j3).

The following dataset was generated:

| Author(s) | Year | Dataset title | Dataset URL | Database and Identifier |
|---|---|---|---|---|
| Baudin J, Angueyra J, Sinha R, Rieke F | 2019 | Data from: S-cone photoreceptors in the primate retina are functionally distinct from L and M cones | http://dx.doi.org/10.5061/dryad.gv5k2j3 | Dryad Digital Repository, 10.5061/dryad.gv5k2j3 |

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
