## [Decision Letter]

Thank you for submitting your manuscript "S cone photoreceptors in the primate retina are functionally distinct from L and M cones" to *eLife*. The manuscript has been reviewed by three peer reviewers, and their assessments, together with those of the Reviewing Editor, form the basis of this letter.

The reviewers and Reviewing Editor think that this body of work fills a niche and that it would be of interest to the human color vision psychophysics community. However, all three reviewers had very substantial questions relating to the data, its analysis, and its presentation that would need to be addressed before we could reconsider the manuscript.

If you would like proceed with submission of a revised manuscript, we ask that you first submit a point-by-point response to the three reviews (appended here, with minor edits). This will allow both you and us to assess the feasibility of that path.

Reviewer #1:

For over half a century, psychophysicists have gathered evidence that the time constants of the short-wave cones of the retina are longer than those of long- and middle-wave cones. So the direct measurements of primate cones now provided by Reike and colleagues are very welcome and are of great interest for understanding human perception. In principle the work is very suitable for publication in *eLife*. I have three primary comments:

1) The paper very much rests on the estimates of isomerization rates in the different types of cone. In paragraph one of subsection “Light stimulation”, the authors say that they assumed an effective collecting area of 0.37 μm2 – for all cones. Yet short-wave cones are reported to be smaller, and certainly, since smaller cones were deliberately selected for testing, there must be a difference in collecting area in those measured. Isn't this a problem?

2) Clinical psychophysics suggest that short-wave cones are less robust than other cones. One can imagine that time constants increase and noise levels increase for all cones the longer the interval between removal of the eye and the moment of recording. An analysis of the authors' existing records might offer a check on whether the properties of short-wave cones simply deteriorate more rapidly.

3) At the end of the Materials and methods and in the Supplementary Material, there's a long section on modelling wavelength discrimination. This detracts from the primary material and would be best excluded. There's a large literature, empirical and theoretical, on wavelength discrimination, and we know that post-receptoral effects are probably at least as important as receptoral factors. A separate modelling paper might recommend itself to the authors.

If the authors would like to show relevance to perception in the present paper, it might be more appropriate to consider something more fundamental (which goes unmentioned in the paper): the larger psychophysical Weber fraction for short-wave cones. See Wyszecki and Stiles, 2nd ed, Table 1(7.4.1).

Subsection “S cone identification”. Doesn't this go back to Anhelt, Kolb and Pflug, 1987?

In reference to the psychophysical literature, it might be worth citing, and considering, the rather good paper by Wisowaty and Boynton (1980), Vision Research. During that era, there were many demonstrations that sensitivity to signals originating in the short-wave cones (including time constants) are influenced by signals originating in the L and M cones.

It would be handy for the reader to have a rough conversion from isomerization rates to trolands.

Reviewer #2:

In this manuscript, the authors add their results and conclusions to an old but perplexing issue which is the comparably slow speed of the blue sensitive channel in the visual system relative to the L/M channel. Psychophysically it is well known that flicker sensitivity is slower in the blue, but it remains to be shown specifically wherein the retinal or cortical circuits the temporal fidelity is limited. This issue is attacked along many vectors in the manuscript, and overall the efforts are very solid. The noise analysed in S-cone behavior is excellent. Because the results on S-cones are few in the published records, it is a pity these authors have not done more to expound on their unique resource.

The main issues with the papers are the cursory review and presentation of their results and lack of detailed information by which to judge their data set. The authors purport to be presenting a new large data set on S cone responses and only in the supplementary figures do we see it is 28 or 18 cells. At the minimum, we must see the families of responses after those of Cao et al. and Baylor and colleagues. The exact filtering and averaging processing of the linear range responses needs to be clear and less of it would be better. It is not clear if the results shown are average linear range responses, or average modelled fits. Showing the averaged normalized linear filters is disloyal and too many steps removed from the actual data.

1) None of the data collected is in dark adapted tissues whereas most of the published comparators are dark adapted. Cannot these tissues be bathed in chromophore to reinstate a dark adapted conditions

a) The noise recordings show dark and low light responses, were and flashes or flash families recorded under these conditions?

b) Were the antibody directed S-cone recordings excluded from the summaries?

2) Show families of responses for S-cone and L vs. M cones. Do the S-cone time-to-peaks shift in a similar manner with increases in stimulus strength?

a) Was the undershoot in the recovery phase ever-present or the same size in S vs. L/M cones? (a zero line in Figure 1A, B, E and F; Figure 1—figure supplement 2A and others would help here).

3) What were the Rmax values of the cone populations, and the variability? Were they similar with respect to spectral type?

4) Perform a phototransduction gain analysis to measure the rising phase of the S vs LM cones and ascertain whether it is the activation or inactivation phases of the responses which are driving the overall slower responses.

5) A measure of the recovery rate, or full-width at half height of the linear range responses. This information would speak to the rate of recovery of the responses as well to the variability of the populations. Integration times of the cone population responses is another important characteristic left out.

a) Did the cone recordings change in their kinetic responses over time? If so, in what direction and how much? Where the changes noted in S-cones similar to those in the M/L cone recordings?

6) In your supplementary figures show the readers the fastest S-cone and the slowest M-L cone response. What were the selection criteria applied for inclusion to the data set? What was the filtering applied to these signals? Since the s-cone sample is large why not show a group of the "fastest 6" and slowest 6 responses as individual traces for each spectral type of cone.

7) Cao et al. demonstrate a calcium dependent undershoot in cone responses suggesting an altered physiological status of isolated cells and tissues. If the S-cones are indeed a fragile cone then it is not a great leap to consider the isolation of these cells in the dish could have a greater impact the S-cone population. Thus, your inclusion/exclusion criteria are of import, as are judgements of the quality and duration of the recordings. Did an undershoot appear in the basal state (not dark adapted) of your recordings over time.?

8) The results of the in vitro ganglion cell recordings and the non-linear kinetic models of the cone responses here should be more carefully compared to the kinetics of the driving forces in for the SBC and P and M ganglion cells in the 2007 Field et al. paper.

*Reviewer #3:*

Baudin and colleagues measure cone photoreceptor and blue-on ganglion cell responses in an intact choroid-detached preparation of macaque monkey retina. A three channel LED stimulator is used. Responses to flashes and sinewave temporal modulation are presented. Ideal observer modelling is also presented. The data appear to have been competently collected and mostly accurately analysed following many similar and excellent productions from the Rieke lab. The main result is that under many conditions S cones lag LM cones, for example by 8.5 ms at 5000 isomerizations/s; consistent results from the periodic stimulus are shown. The S cones also show more intrinsic noise at low light levels. The cone latency difference appears swamped by synaptic circuitry on the way to blue-on ganglion cells, but the basic demonstration of delay should be of interest within the field.

There appear to be some useful observations here but the manuscript and results are so tendentiously treated and difficult to follow that one is left feeling more puzzled at the end of reading than at the beginning.

Some examples follow.

1) Abstract: "Models for cone vision often assume that these three cone

11 types are functionally identical aside from differences in their spectral sensitivity." This and related statements made throughout the manuscript do not hold water. Here, for example, is an authoritative and prominent statement (last sentence of Abstract) from the leaders of the field in S cone psychophysics (Lee et al., 2009): "The data imply that the S-cone signal is delayed by approximately 12 ms.… we suggest that the delay arises before the S-opponent channels are constructed, possibly in the S-cones themselves." The physiological and psychophysical literature is replete with these sorts of statements. The demonstration here that S cones are under many conditions slower than M and L cones is an interesting and useful confirmation of what many (probably most) workers have assumed. Writing as if the result is a paradigm-shifter is not very helpful.

2) Figure 1C, D, other figure panels and associated text. It seems very strange to be presented with scatterplots where each point in fact shows two (?) independent (?) observations from unstated numbers of repeat (?) measurements on two different cones. Without information on how many cones were sampled in each piece of retina, and how an example of each class was (one hopes) randomly(?) selected for comparison it is all very opaque. The Materials and methods do not even tell us how many monkeys were used, how many pieces were sampled, how many pieces contribute how many points to each scatterplot. Animals could be distinguished by different symbols for example. Readers also deserve an introductory figure showing raw examples or at very least some inset traces to get a feel for the quality of the data, please.

3) The reader needs to work through to Figure 2 to find that at 1000 R*/s the S and M curves show perfect overlap. This is interesting but is not addressed squarely, and is papered over by the normalized latency plots. All the information presented to this point has been on higher intensities where there is a difference in latency, with no forward reference to intensity-dependence.

4) 1000 R*/s would correspond roughly to high mesopic (indoor lighting) levels, yes? As rods are not clamped by the three-channel stimulator the reader deserves data indicating lack of rod intrusion to the cone and ganglion cell recordings. Forgive me if I have missed something, but the manuscript seems to be written as if recordings were made from isolated cones, not cones embedded in (hopefully) intact retinal circuitry.

5) Subsection “Differences in cone response properties shape the retinal output”: "Directly testing this prediction is difficult due to a lack of efficiently targetable ganglion cells that receive S and LM cone signals via identical pathways." This statement is very strange: the retina is the retina, not a computer chip. The following descriptions and analysis in this section is circumlocutory. The ganglion cell S cone kinetics appear faster than the ML cone kinetics, this result is not consistent with previous reports from retina (macaque) or LGN (macaque, marmoset). The discrepancy needs to be addressed squarely please and the latency measures presented.

6) The ideal observer analysis is not even mentioned in the results and seems quite disconnected from the data.

7) Many passages are presented with only reference to figure parts and no numerical support in the text. Paras L57ff and L179ff are glaring example but there are many others.

8) For the authors consideration. The delay should be manifest as phase shift in the Fourier transformed data, yes? The analysis appears to take only amplitude spectra.

9) Subsection "Noise in S cones is unique from that in LM cones" [meaning unclear].

10) Regarding n=18 cones, I make out only 14 data points in the scatterplot in Figure 1—figure supplement 2, perhaps there were some piece(s?) where an S cone but no M/L cone was measured.

---

## [Author Response]

Reviewer #1:

For over half a century, psychophysicists have gathered evidence that the time constants of the short-wave cones of the retina are longer than those of long- and middle-wave cones. So the direct measurements of primate cones now provided by Reike and colleagues are very welcome and are of great interest for understanding human perception. In principle the work is very suitable for publication in eLife. I have three primary comments:1) The paper very much rests on the estimates of isomerization rates in the different types of cone. In paragraph one of subsection “Light stimulation”, the authors say that they assumed an effective collecting area of 0.37 μm2 – for all cones. Yet short-wave cones are reported to be smaller, and certainly, since smaller cones were deliberately selected for testing, there must be a difference in collecting area in those measured. Isn't this a problem?

Thank you for highlighting this issue. Collecting areas could indeed differ for S vs L/M cones, but we believe that such differences are unlikely to account for our results for several reasons. First, the differences in collecting area are expected to be relatively small. S cone responses at 50,000 R*/s are slower than those of L/M cones at 5,000 R*/s; hence, to account for our results, collecting areas would need to differ at least than 10-fold. Second, the difference in cone collecting areas is expected to be small in the fovea (Anhelt et al., 1987), but the differences in S vs L/M cone kinetics persists in foveal cones. Third, focusing of light through the inner segment does not appear to make a large contribution to the collecting area in our preparations – i.e. collecting areas estimated for light that traverses or does not traverse the cone inner segment were similar (as now noted in the Materials and methods). We have added a discussion of the potential difference in collecting area and why we think it does not have a large impact on interpretation of our results (subsection “Light stimulation”).

2) Clinical psychophysics suggest that short-wave cones are less robust than other cones. One can imagine that time constants increase and noise levels increase for all cones the longer the interval between removal of the eye and the moment of recording. An analysis of the authors' existing records might offer a check on whether the properties of short-wave cones simply deteriorate more rapidly.

This is another good point. Our recordings were made from 2 to 15 hours after collecting the retinas. We have compared kinetics across this time period and find that the difference in S vs L/M kinetics persists in different time windows throughout this range (i.e. 2-6 hours, 6-10 hours, 10-15 hours). Overall, we see at most small differences in the kinetics of cone responses or in noise over time, although we only test cones that have maximal responses that meet our criteria for inclusion. We have added information about the consistency of the slower S cone responses over time as part of new “Cell-selection criteria” section in Materials and methods.

One aspect of the cone responses that can change (particularly decline) over time is the maximal light response (and hence the proportion of recorded cells from which we collect data). However, we use the response amplitude to select cells to record from, with the aim of selecting against cells with small, potentially abnormal, light responses. The assumption underlying this selection is that the largest responses we observe most closely resemble those in vivo, and have added this assumption to the Materials and methods.

3) At the end of the Materials and methods and in the Supplementary Material, there's a long section on modelling wavelength discrimination. This detracts from the primary material and would be best excluded. There's a large literature, empirical and theoretical, on wavelength discrimination, and we know that post-receptoral effects are probably at least as important as receptoral factors. A separate modelling paper might recommend itself to the authors.If the authors would like to show relevance to perception in the present paper, it might be more appropriate to consider something more fundamental (which goes unmentioned in the paper): the larger psychophysical Weber fraction for short-wave cones. See Wyszecki and Stiles, 2nd ed, Table 1(7.4.1).

The reviewer’s comments on the wavelength discrimination model are well taken and we have removed that section from the paper.

Subsection “S cone identification”. Doesn't this go back to Anhelt, Kolb and Pflug, 1987?

Yes – we have added the above reference.

In reference to the psychophysical literature, it might be worth citing, and considering, the rather good paper by Wisowaty and Boynton (1980), Vision Research. During that era, there were many demonstrations that sensitivity to signals originating in the short-wave cones (including time constants) are influenced by signals originating in the L and M cones.

Thank you for pointing out that study. We have added a couple sentences about the perceptual interactions between L/M cones and S cones to the Discussion as part of a (largely new) discussion of the relationship between our results and perceptual results. This discussion focuses on the importance of post-receptoral interactions, which seem likely to account for the Wisowaty and Boyton and related results.

It would be handy for the reader to have a rough conversion from isomerization rates to trolands.

Thanks – we have added that conversion to the Materials and methods where we describe the stimuli.

Reviewer #2:

In this manuscript, the authors add their results and conclusions to an old but perplexing issue which is the comparably slow speed of the blue sensitive channel in the visual system relative to the L/M channel. Psychophysically it is well known that flicker sensitivity is slower in the blue, but it remains to be shown specifically wherein the retinal or cortical circuits the temporal fidelity is limited. This issue is attacked along many vectors in the manuscript, and overall the efforts are very solid. The noise analysed in S-cone behavior is excellent. Because the results on S-cones are few in the published records, it is a pity these authors have not done more to expound on their unique resource.The main issues with the papers are the cursory review and presentation of their results and lack of detailed information by which to judge their data set. The authors purport to be presenting a new large data set on S cone responses and only in the supplementary figures do we see it is 28 or 18 cells. At the minimum, we must see the families of responses after those of Cao et al. and Baylor and colleagues. The exact filtering and averaging processing of the linear range responses needs to be clear and less of it would be better. It is not clear if the results shown are average linear range responses, or average modelled fits. Showing the averaged normalized linear filters is disloyal and too many steps removed from the actual data.

Thank you for this feedback. We have added example cells throughout the paper. This includes: (1) a new Figure 1 showing all recorded flash responses from peripheral cones; (2) example L, M and S cones in Figure 3; and, (3) adding example L, M and S cone responses to noise stimuli and showing filters from all recorded cells in Figure 4.

The displayed traces are minimally filtered; we record the original data with a 3 kHz bandwidth and do not apply any additional filtering to the displayed traces, except that responses to noise stimuli in Figure 4A. We have clarified this in the Materials and methods. All of the traces shown are averaged recorded responses rather than model fits; the linear filters in Figure 4 (previously Figure 1—figure supplement 2) are computed directly from measured responses to Gaussian-noise stimuli. We have added information about the number of recorded cells for each experiment in the main text, including the total number of recorded cells of each type (subsection “Cell-selection criteria”).

Because our recordings are quite short in duration (3-5 minutes maximum), we do not record full flash families. We do have some short flash families (see response to item (1) below).

1) None of the data collected is in dark adapted tissues whereas most of the published comparators are dark adapted. Cannot these tissues be bathed in chromophore to reinstate a dark adapted conditions

All of our recordings start from fully dark-adapted retinas. After isolating the eyecup, we dark-adapt the retina attached to the pigment epithelium for a period of at least 1 hour before starting to record. This is sufficient time to achieve complete dark adaptation, as indicated by the ability to measure single photon responses from rods and downstream retinal ganglion cells (e.g. Ala-Laurila and Rieke, 2014). We initiate our cone recordings from darkness and then measure responses at different background luminances. We chose to emphasize cone responses in the presence of a background rather than from darkness both to saturate rods (see reviewer 3 point 4) and to more closely replicate the operation of cones under photopic conditions. Following a successful recording from a peripheral cone, we move to a location on the retina that was outside the region exposed to light before attempting another recording; this ensures that all recorded cells are fully dark adapted at the start of the recording. Given the small size of the fovea we were limited to fewer new regions after recording from a foveal cone. We have clarified these issues in the Materials and methods.

a) The noise recordings show dark and low light responses, were and flashes or flash families recorded under these conditions?

Noise recordings were based on light steps (3 seconds) from darkness to different background luminance. We have added a description of the exact stimulus protocol to the Materials and methods. We also recorded flash responses at each background at which noise was measured in a subset of the cells. We did not record full flash families because of the time required to make noise measurements and the short duration over which we could be confident that the light response was stable.

b) Were the antibody directed S-cone recordings excluded from the summaries?

None of the data reported is from cells targeted with antibodies. We have added that to the Materials and methods.

2) Show families of responses for S-cone and L vs. M cones. Do the S-cone time-to-peaks shift in a similar manner with increases in stimulus strength?

As mentioned above, we did not record flash families from most cells due to the limited duration of our experiments. We did record families to a few contrasts in some cases. An example flash family (over a limited range of contrasts) and analysis of the dependence of the time-to-peak on flash strength are shown below; we opted not to include this figure in the paper, but are happy to do so if the reviewer feels that it is important. See Author response image 1.

a) Was the undershoot in the recovery phase ever-present or the same size in S vs. L/M cones? (a zero line in Figure 1A, B, E and F; Figure 1—figure supplement 2A and others would help here).

The undershoot in the recovery phase did not differ substantially between S vs L/M cones at any light level. We have added zero lines to all figures with flash responses to help identify the undershoot and compare across cells. We have also included a figure (Figure 2—figure supplement 2) in which the time-to-peak of the cone responses are normalized to 1 to facilitate comparison of the response shape, including the undershoot.

3) What were the Rmax values of the cone populations, and the variability? Were they similar with respect to spectral type?

We did not record maximal responses from each cell, but instead proceeded with recording as long as responses exceeded a criterion amplitude. We have now specified those cell selection criteria in a new section in the Materials and methods. We did not observe large differences in maximal responses from different cone types.

4) Perform a phototransduction gain analysis to measure the rising phase of the S vs LM cones and ascertain whether it is the activation or inactivation phases of the responses which are driving the overall slower responses.

Thank you – this is an excellent suggestion. We started by comparing the rising and recovery phases of the responses. To do so, we scaled the responses to a time-to-peak of one and a peak amplitude of one (Figure 2—figure supplement 2). With this scaling, the response shapes were quite similar. This indicates that the slowing of the S cone response is not restricted to the rising or recovery phases, which we now note in the Results (subsection “S-cone responses are slower than those of L- and M- cones”, fifth paragraph) and in the Discussion (subsection “Possible mechanistic origins of differences in cone signals”, second paragraph). We did not feel that the phototransduction gain analysis added additional information since it measures the initial slope of the response.

5) A measure of the recovery rate, or full-width at half height of the linear range responses. This information would speak to the rate of recovery of the responses as well to the variability of the populations. Integration times of the cone population responses is another important characteristic left out.

Changes in full-width at half max were very similar to changes in the time to peak. This is clear from the (new) Figure 2—figure supplement 2, which shows that the responses across cone types superimpose closely when their time axes are normalized. We also added results from the full-width-at-half-max analysis to Figure 2C and D and the main text (subsection “S-cone responses are slower than those of L- and M- cones”, second paragraph).

a) Did the cone recordings change in their kinetic responses over time? If so, in what direction and how much? Where the changes noted in S-cones similar to those in the M/L cone recordings?

The cones that have been included in this study were stable during the short duration of the recording. Stability of kinetics is one of the criteria we used to determine the duration of a recording (we can maintain a good recording from a cell for much longer than 3-4 minutes, but the responses clearly start to change after ~5 min). We have added this information to the Materials and methods in describing the recordings and selection criteria. Also see reviewer 1 point 2 on stability of cone responses at various times after isolation of the retina.

6) In your supplementary figures show the readers the fastest S-cone and the slowest M-L cone response. What were the selection criteria applied for inclusion to the data set? What was the filtering applied to these signals? Since the s-cone sample is large why not show a group of the "fastest 6" and slowest 6 responses as individual traces for each spectral type of cone.

We appreciate the push to show less derived responses, and considered several possibilities for how to do this. We decided that, rather than show a selection of responses, it was better show all the recorded cells in Figures 1 and 4. We felt that this gave an immediate sense of the full data set and the degree of consistency of the responses across cells of a given type.

The selection criteria used for inclusion of data was based on response amplitudes to light flashes at the start of every recording. All cells included in analysis of voltage clamp data had maximal light responses >100 pA, and nearly all cells included in analysis of current clamp data had responses of >10 mV, with 15 mV more typically. Occasionally we retained a cell with a smaller response if the noise was low and sensitivity was otherwise good. We have added a section to the Materials and methods detailing these response criteria. The same criteria were used across cone types.

We filtered the responses during recording at 3 kHz, but did not perform any filtering of flash responses post-recording. Linear filters in Figure 4 were filtered at 60 Hz. We have clarified these issues in the Materials and methods.

7) Cao et al. demonstrate a calcium dependent undershoot in cone responses suggesting an altered physiological status of isolated cells and tissues. If the S-cones are indeed a fragile cone then it is not a great leap to consider the isolation of these cells in the dish could have a greater impact the S-cone population. Thus, your inclusion/exclusion criteria are of import, as are judgements of the quality and duration of the recordings. Did an undershoot appear in the basal state (not dark adapted) of your recordings over time.?

This is an important concern and we have now included several analyses to compare the susceptibility of S cone kinetics and amplitude vs L/M cones over time. First, in response to reviewer 1 point 2, we have included an analysis of response kinetics and amplitude of all our cone recordings as a function of time of recording after collecting the retina (Materials and methods, subsection “Cell-selection criteria”). This analysis shows that the difference in kinetics is present during our earliest recordings. Second, as mentioned above, we now show responses from all peripheral cones in which we recorded flash responses in Figure 1 so the reader can judge the consistency of the responses. We also added a section to the Materials and methods describing our cell selection criteria.

The extracellular calcium concentration of 1.15 mM used in this study is similar to that of Cao et al., 2014 (1.2 mM). The cone responses are largely in accordance with Cao et al., 2014 in that they have only a small undershoot. We never observed undershoots as pronounced as those shown in a few cells in Cao et al., 2014 or those from earlier primate cone recordings. As mentioned above, we do not observe any systematic difference in this undershoot across cone types (see Figure 2—figure supplement 2).

All the recordings in our study were performed in an intact whole mount retina where the photoreceptor array and the underlying retinal circuits are preserved as much as possible. A retinal region that had L/M cones with good light sensitivity also contained S cones that had good light sensitivity – and similarly poor L/M cone responses were predictive of poor S cone responses. To control for systematic differences in the health of the retina from which recordings were made, we referenced all recorded S cones to nearby L/M cones (Figure 2E, F; Figure 4C)

*8) The results of the* in vitro *ganglion cell recordings and the non-linear kinetic models of the cone responses here should be more carefully compared to the kinetics of the driving forces in for the SBC and P and M ganglion cells in the 2007 Field et al. paper.*

In our recordings, the time to peak of the S cone-mediated response at 1000 R*/s is ~40 ms and the time to peak of the L/M response is ~55 ms. Field et al., 2007, report a time to peak for the S cone response of just over 50 ms, and a time to peak of the L+M response of 70 ms; light levels in their experiments were comparable to our low-light condition (i.e. 1000 R*/s). The parasol responses in that study are also somewhat slower to peak than ours. Most importantly, Field et al. observe considerably slower L+M responses than S responses (70 vs 50 ms), and this is consistent with the extra synapse required for the L+M responses. We note this comparison now in subsection “Differences in cone response properties shape the retinal output”, paragraph four. We have also added the absolute numbers for the time to peak of the S and LM SBC filters to the text.

Reviewer #3:

Baudin and colleagues measure cone photoreceptor and blue-on ganglion cell responses in an intact choroid-detached preparation of macaque monkey retina. A three channel LED stimulator is used. Responses to flashes and sinewave temporal modulation are presented. Ideal observer modelling is also presented. The data appear to have been competently collected and mostly accurately analysed following many similar and excellent productions from the Rieke lab. The main result is that under many conditions S cones lag LM cones, for example by 8.5 ms at 5000 isomerizations/s; consistent results from the periodic stimulus are shown. The S cones also show more intrinsic noise at low light levels. The cone latency difference appears swamped by synaptic circuitry on the way to blue-on ganglion cells, but the basic demonstration of delay should be of interest within the field.There appear to be some useful observations here but the manuscript and results are so tendentiously treated and difficult to follow that one is left feeling more puzzled at the end of reading than at the beginning.Some examples follow.1) Abstract: "Models for cone vision often assume that these three cone11 types are functionally identical aside from differences in their spectral sensitivity." This and related statements made throughout the manuscript do not hold water. Here, for example, is an authoritative and prominent statement (last sentence of Abstract) from the leaders of the field in S cone psychophysics (Lee et al., 2009): "The data imply that the S-cone signal is delayed by approximately 12 ms.… we suggest that the delay arises before the S-opponent channels are constructed, possibly in the S-cones themselves." The physiological and psychophysical literature is replete with these sorts of statements. The demonstration here that S cones are under many conditions slower than M and L cones is an interesting and useful confirmation of what many (probably most) workers have assumed. Writing as if the result is a paradigm-shifter is not very helpful.

Thank you for the constructive feedback. We have replaced the statement about functional similarity of cones with a short summary of the physiological and psychophysical literature (first paragraph of the Introduction). This paragraph mentions previous studies that demonstrated differences or delays in S cone vs L/M signals in the retina (Field et al., 2007), LGN (Tailby et al., 2008) and visual cortex (Cottaris and De Valois, 1998). In addition, we mention S cone psychophysical studies that have observed slower responses to S cone isolating stimuli (Lee et al., 2009; Smithson and Mollon 2004; Shinomori and Werner 2008; Stromeyer et al., 1991). As the reviewer points out, and as mentioned in the above reference (Lee et al. 2009), although the delay in S cone signals was hypothesized to originate in the S cones, previous recordings from S cones have not revealed a clear difference in kinetics (Schnapf et al., 1990). We also return to the issue of the relation of our findings with psychophysics in the Discussion. We hope that these changes provide a fairer and more complete treatment of how our work fits in with previous findings.

2) Figure 1 C, D, other figure panels and associated text. It seems very strange to be presented with scatterplots where each point in fact shows two (?) independent (?) observations from unstated numbers of repeat (?) measurements on two different cones. Without information on how many cones were sampled in each piece of retina, and how an example of each class was (one hopes) randomly(?) selected for comparison it is all very opaque. The Materials and methods do not even tell us how many monkeys were used, how many pieces were sampled, how many pieces contribute how many points to each scatterplot. Animals could be distinguished by different symbols for example. Readers also deserve an introductory figure showing raw examples or at very least some inset traces to get a feel for the quality of the data, please.

Thank you – our description of those plots was not sufficiently clear. To help control for possible bias in comparing S vs L/M cone responses across retinas, we chose to compare the mean time to peak of S cones from a single piece of retina to the mean time to peak of L and M cones in the same retinal piece. Each point in the scatter plots reflects these mean times to peak; no selection was made of which cells to include, except the general cell-selection criteria mentioned now in the Materials and methods (which focus on response amplitude). We have revised the text to clarify this comparison. We have also included details about the number of retinal pieces from which data was collected (>10 for most data; mentioned in the figure legend) as well as the total number of cells that contribute to each panel. The Materials and methods now include information about the total number of recorded cells across experiments and the minimum number of animals.

The suggestion of an introductory figure that orients a reader to the data is a very good one. We followed this by adding a figure (Figure 1) that shows flash responses from each measured cone along with averages across the lighting conditions studied.

3) The reader needs to work through to Figure 2 to find that at 1000 R*/s the S and M curves show perfect overlap. This is interesting but is not addressed squarely, and is papered over by the normalized latency plots. All the information presented to this point has been on higher intensities where there is a difference in latency, with no forward reference to intensity-dependence.

That is a good point. We now start the paper with a new figure (following the suggestion above) that provides a complete view of the flash response data set (Figure 1). This figure includes averages from different cone types at each mean light level, and averages within a cone type across light levels. This allows us to anticipate the difference in kinetics and the greater dependence of kinetics of LM cone responses on background (subsection “S-cone responses are slower than those of L- and M- cones”). We refer back to this figure to directly note the similarity of the kinetics across cone types at 1,000 R*/cone/s.

4) 1000 R*/s would correspond roughly to high mesopic (indoor lighting) levels, yes? As rods are not clamped by the three-channel stimulator the reader deserves data indicating lack of rod intrusion to the cone and ganglion cell recordings. Forgive me if I have missed something, but the manuscript seems to be written as if recordings were made from isolated cones, not cones embedded in (hopefully) intact retinal circuitry.

Experiments for a separate project specifically on mesopic conditions indicate that the upper end of mesopic vision is closer to 300-500 R*/s, at least under our experimental conditions. This conclusion comes from direct measurements in rods and spectral measurements in RGCs. We now specifically mention this conclusion and reference the paper (Grimes et al., *eLife*, 2018) that it is based on in the Materials and methods. We have also clarified that all of the recordings in the paper were made from cones in intact retinas rather than from isolated cones.

5) Subsection “Differences in cone response properties shape the retinal output”: "Directly testing this prediction is difficult due to a lack of efficiently targetable ganglion cells that receive S and LM cone signals via identical pathways." This statement is very strange: the retina is the retina, not a computer chip. The following descriptions and analysis in this section is circumlocutory. The ganglion cell S cone kinetics appear faster than the ML cone kinetics, this result is not consistent with previous reports from retina (macaque) or LGN (macaque, marmoset). The discrepancy needs to be addressed squarely please and the latency measures presented.

As mentioned by the reviewer, latencies to cone isolating stimuli have been measured at several stages of the visual pathway i.e. in the retina, LGN and visual cortex. In the retina, faster S cone kinetics compared to L/M cone kinetics have been seen previously in small bistratified ganglion cells at comparable light levels (Field et al., 2007). This has been attributed to L/M cone signals traversing an additional synapse on their way to SBCs. This is why we refrained from a direct comparison of the kinetics. Therefore, we compared another distinguishing feature of the S cone signals -- the weak dependence of S cone response kinetics on background compared to that of LM cones. This is what we tested in the SBC ganglion cells and not the absolute kinetic differences. We have revised the introduction to the section on SBCs to help clarify these issues (subsection “Differences in cone response properties shape the retinal output”), and mention the similarity of the absolute kinetics of the responses we observe to those measured previously at comparable light levels (paragraph four).

Previous work at higher light levels did not see the difference in kinetics of S vs LM responses that we observe and reported in Field et al., 2007. This is consistent with our observations that the difference in kinetics decreases with increasing light level.

6) The ideal observer analysis is not even mentioned in the results and seems quite disconnected from the data.

We removed this analysis from the paper based on this comment and similar concerns from the first reviewer.

7) Many passages are presented with only reference to figure parts and no numerical support in the text. Paras L57ff and L179ff are glaring example but there are many others.8) For the authors consideration. The delay should be manifest as phase shift in the Fourier transformed data, yes? The analysis appears to take only amplitude spectra.

We originally relegated most of the statistical analyses to the figure captions in the original paper. We have added statistics for key points to the main text (see Results section).

Thank you – we looked at the phase spectra but found them difficult to interpret. This may in part be because the difference in kinetics is not a straight delay but a slowing of the entire response (see new Figure 2—figure supplement 2 which normalizes the time-to-peak). Because the phase plots were difficult to interpret we have not included them in the paper, but they are below in case the reviewer is interested or sees something in them that we missed. See Author response image 2.

**Author response image 2. respfig2:** 

9) Subsection "Noise in S cones is unique from that in LM cones" [meaning unclear].

Thank you – we changed that section title to “Noise in S cones differs from that in L/M cones”.

10) Regarding n=18 cones, I make out only 14 data points in the scatterplot in Figure 1—figure supplement 2, perhaps there were some piece(s?) where an S cone but no M/L cone was measured.

That is correct – for some S cones we did not have a matched recording from a L or M cone. We show all S cones, and all L and M cones, from which we measured filters using Gaussian noise in Figure 4B (previously Figure 1—figure supplement 2). A subset of these cones appear in Figures 4C and D. We now note that in the figure caption.